

# First Investigation of Perennial Ice in Winter Wonderland Cave, Uinta Mountains, Utah, USA

Jeffrey S. Munroe[1]

[1]Geology Department, Middlebury College, Middlebury, VT, 05753, USA

*Correspondence to*: Jeffrey S. Munroe (jmunroe@middlebury.edu)

**Abstract.** Winter Wonderland Cave is a solution cave at an elevation of 3140 m above sea level in Carboniferous-age Madison Limestone on the southern slope of the Uinta Mountains (Utah, USA). Temperature dataloggers reveal that the mean annual air temperature (MAAT) in the main part of the cave is -0.8°C, whereas the entrance chamber has a MAAT of -2.3°C. The MAAT outside the cave entrance was +2.8°C between August 2016 and August 2018. Temperature in excess of 0°C were not recorded

inside the cave during that 2-year interval. About half of the accessible cave, which has a mapped length of 245 m, is floored by perennial ice. Field and laboratory investigations were conducted to determine the age and origin of this ice and its possible paleoclimate significance. Ground penetrating radar surveys with a 400-MHz antenna reveal that the ice has a maximum thickness of ~3 m. Samples of packrat (*Neotoma*) droppings obtained from the ice in the main part of the cave yielded radiocarbon ages from 40 ± 30 to 285 ± 12 years. These results correspond with median calibrated ages from AD 1645 to 1865,

suggesting that most of the ice accumulated during the Little Ice Age. Samples collected from a ~2-m high exposure of layered ice were analysed for stable isotopes and glaciochemistry. Most values of $\delta^{18}O$ and $\delta D$ range plot subparallel to the global meteoric water line with a slope of 7.5 with an intercept of 0.03‰. Values from some individual layers depart from this local water line suggesting that they formed during close-system freezing. In general, values of both $\delta^{18}O$ and $\delta D$ are lowest in the deepest ice, and highest at the top. This trend is interpreted as a shift in the relative abundance of depleted winter precipitation

and enriched summer precipitation over time. Calcium has the highest average abundance of cations detectable in the ice (mean of 6050 ppb), followed by Al (2270 ppb), Mg (830 ppb), and K (690 ppb). Most elements are more abundant in the younger ice, possibly reflecting reduced rates of infiltration that prolonged water-rock contact in the epikarst. Abundances of Al and Ni likely reflect eolian dust incorporated in the ice. Liquid water appeared in the cave in August 2018 and August 2019, apparently for the first time in many years. This could be a sign of a significant change in the cave environment.

## 1 Introduction

Caves containing perennial ice, hereafter known as "ice caves", have been reported from around the world (Persoiu and Lauritzen, 2017). Local permafrost conditions are maintained in these caves due to a combination of cold air trapping and dynamic ventilation (Luetscher and Jeannin, 2004a). Ice in these caves is a product of firnification of snow that falls or slides into the entrance, congelation of water entering into the cave, or both. Previous studies have demonstrated the paleoclimate

potential of the perennial ice in caves (Holmlund et al., 2005; Perşoiu et al., 2017), which can be investigated with methods analogous to those applied to glacier ice cores (Yonge, 2014; Yonge and MacDonald, 1999). Significantly, because ice caves are present at lower latitudes and altitudes than many mountain glaciers, they provide an opportunity to expand cryosphere-based paleoclimate records to areas where surficial ice is absent (Perşoiu and Onac, 2012). An important motivation for increased study of ice caves is the observation that many perennial ice bodies in these caves are currently melting (Fuhrmann, 2007;



Pflitsch et al., 2016). This worrisome trend raises the alarm that the paleoclimate records preserved in these subterranean ice masses will be lost forever (Kern and Perşoiu, 2013; Veni et al., 2014).

Despite the exciting potential of ice caves as paleoclimate archives, these features remain an understudied component of the cryosphere. Although an overview of a broad array of ice caves in a wide range of settings was presented more than 100 years

ago (Balch, 1900), most modern investigations of cave ice have been conducted in a few focused areas, particularly the Alps (e.g. Luetscher et al., 2005; May et al., 2011; Morard et al., 2010) and the mountains of Romania (e.g. Perşoiu et al., 2011; Persoiu and Pazdur, 2011). There, techniques for dating cave ice (e.g. Kern et al., 2009; Luetscher et al., 2007; Spötl et al., 2014), interpreting stable isotope records in cave ice (e.g. Kern et al., 2011b; Perşoiu et al., 2011), and studying the glaciochemistry (e.g. Carey et al., 2019; Kern et al., 2011a) of cave ice have been developed over the past several decades.

Technologies such as environmental dataloggers have simultaneously enabled investigation of micrometeorology in ice caves (e.g. Luetscher and Jeannin, 2004b; Obleitner and Spötl, 2011), and geophysical techniques such as ground penetrating radar (GPR) have been employed to image ice stratigraphy (e.g. Behm and Hausmann, 2007; Colucci et al., 2016; Gómez Lende et al., 2016; Hausmann and Behm, 2010, 2011). In general, however, these techniques have been employed only on a limited basis elsewhere in the world. For instance, in North America, the ablation of subterranean ice masses has been monitored in New

Mexico (Dickfoss et al., 1997) as well as in lava tubes on the Big Island of Hawaii (Pflitsch et al., 2016) and in California (Fuhrmann, 2007). Seminal work on stable isotopes in cave ice was conducted in the Canadian Rocky Mountains (Lauriol and Clark, 1993; Yonge, 2014; Yonge and MacDonald, 1999), and micrometeorology studies have been published on sites with (possibly) perennial ice in the northern Appalachian Mountains (Edenborn et al., 2012; Holmgren et al., 2017).

The most extensive investigation of a North America ice cave focused on Strickler Cavern, a site in the Lost River Range of Idaho (Munroe et al., 2018). That work documented coexistence of firn ice and congelation ice with radiocarbon age control extending back ~2000 years. Stable isotopes in this ice were interpreted to record cooling temperatures leading into the Little Ice Age, and analysis of major and trace elements supported identification of a local component and an exotic component of the overall dissolved load in the ice.


The project presented here applied the successful multidisciplinary approach from Strickler Cavern (Munroe et al., 2018) to another cave containing ice of a different genesis. In addition to radiocarbon dating, and glaciochemical and isotopic analysis, a two-year deployment of temperature dataloggers was incorporated to constrain cave meteorology, and GPR was employed to determine the thickness of the perennial ice body. The primary objectives of the project were to:

1)      Determine the origin, extent, and age of the ice in this cave;

2)      Develop and interpret a stable isotope record for this ice;

3)      Analyse and interpret the glaciochemistry of the ice

## 2 Methods

### 2.1 Field site

Fieldwork was conducted within Winter Wonderland Cave in the Uinta Mountains of northeastern Utah, USA. Winter Wonderland Cave (hereafter "WWC") was discovered in 2012 and has an entrance at an elevation of 3140 m on a north-facing cliff of Carboniferous-age Madison Limestone. The general location is presented in Figure 1, but to preserve this fragile cave



environment and the ice formations within it, the exact coordinates of the cave entrance are intentionally withheld. The cave has a vertical extent of 33 m and a mapped length of 245 m (Fig. 1c). The narrow entrance slopes down steeply ~10 m to a roughly

circular room (~10 m) with a high ceiling and flat floor of ice. From this "Icicle Room" a narrow crack leads off to intersect with the "Frozen Freeway", a much larger passage that forms the main part of the cave. This steep-sided canyon, with walls up to 8 m high, developed along a fracture bearing 160/340°. About 50% of the floor of the Frozen Freeway is ice, the remainder is breakdown from the cave ceiling (Fig. 1c). The Frozen Freeway terminates in a wider section with a flat ice floor called the "Skating Rink". Beyond the Skating Rink, the cave splits into two passages, both of which are choked with rock debris. Strong

air currents passing through these rock piles suggest the presence of considerable additional passage beyond the chokes. This air current has sublimated the ice at the edge of the Skating Rink, producing an exposure ~2 m tall (Fig. 1c).

Fieldwork focused on documenting the temperature conditions within WWC, determining the thickness of the perennial ice, gathering organic remains for radiocarbon dating, and collecting ice samples for glaciochemical and isotopic analysis. Visits to

WWC for this project were made on August 11, 2016; August 29, 2018; and August 19, 2019.

### 2.2 Cave Temperature Monitoring

Three temperature dataloggers (Onset Temp Pro v2) were deployed in within the cave from August, 2016 through August, 2018. Each logger was suspended from the ceiling to measure temperature of the free air away from the rock walls and ice surface.

The loggers were set to record the temperature every hour. An additional logger was deployed within a solar radiation shield outside the cave entrance to record ambient air temperatures. The temperature dataloggers were collected in late August, 2018. Data were downloaded into the proprietary Onset software Hoboware, filtered to calculate daily mean values, and exported to a spreadsheet for further analysis and plotting.

### 95    2.3 Ground Penetrating Radar

To constrain the thickness of the ice, GPR surveys were conducted along the Frozen Freeway in 2019 following standard protocols (e.g. Hausmann and Behm, 2011). A shielded 400-MHz antenna and a GSSI SIR-3000 controller were used to collect the radar data. Before the survey, a transect was established with measured points every 5 m. A mark was recorded in the GPR data each time the antenna passed one of these points, which allowed the GPR results to be converted to a physical horizontal

scale. A shorter survey was also conducted in the Icicle Room. In both locations, surveys were conducted multiple times, in both directions, using a variety of gain settings to maximize the balance between detecting stratigraphy in the ice and simultaneously imaging the underlying ice-rock interface.

Data collected with the GPR system were processed with the proprietary GSSI software Radan 7.0. Processing involved

standard steps (e.g. Colucci et al., 2016; Hausmann and Behm, 2010, 2011) including a time-zero correction to eliminate the impulse passing directly from the antenna to receiver (the direct wave); a full-pass background removal to remove the surface wave; a finite impulse response (FIR) stacking filter to remove airwave reflections from the cave walls; an additional FIR filter to clip the bandwidth between 300 and 700 MHz; an adaptive gain procedure to amplify faint, deeper reflectors; distance normalization to produce a horizontally scaled profile; and migration to remove hyperbolic reflectors produced by objects within

the ice. A dielectric permittivity of 3.15, typical of pure ice (Thomson et al., 2012), was used to convert two-way radar travel



times to estimated depths below the surface within the ice body. Because the focus was on determining the thickness of the ice, no attempt was made to account for the different permittivity of the underlying bedrock.

### 2.4 Geochronology

To constrain the age of the ice, packrat (*Neotoma*) droppings within the ice were retrieved by angling an ice screw so that it intersected with the dropping and raised it to the surface. This technique was limited to droppings visible within the upper 15 cm of the ice, corresponding to the length of the ice screw. However, because the surface of the ice was locally sublimated into a series of valleys and troughs with relief of ~30 cm, drilling in the base of the troughs permitted the retrieval of droppings from deeper levels beneath the original ice surface. Samples were collected along the length of the Frozen Freeway, as well as from

the Icicle Room.

Samples of packrat droppings were selected for AMS $^{14}$C dating at NOSAMS (National Ocean Sciences Accelerator Mass Spectrometry facility) and ICA (International Chemical Analysis Inc.). Each was dried, weighed, and photographed before submission. Resulting radiocarbon ages were calibrated against the IntCal 13 calibration curve data (Reimer et al., 2013) in

Oxcal 4.3. All of the samples yielded multiple possible calibration ranges, so the median of the 2-sigma calibration range is considered the most likely age for these samples.

### 2.5 Isotopic and Glaciochemical Analyses

Ice samples were collected from existing exposures using an approach similar to that employed in previous studies (e.g. Carey et

al., 2019). Using a hand-operated ice screw, a total of 84 samples were collected from the 2-m-high exposure of ice at the edge of the Skating Rink. Working downward from the top, samples were collected at a spacing of 2 cm for the first 150 cm, and 5 cm for the bottom 50 cm. At each level, two holes were drilled side-by-side and the resulting miniature ice cores were collected in 50-ml screwtop tubes. The first several centimetres of ice drilled out at each hole were discarded to avoid the possibility of meltwater contamination near the ice surface (Kern et al., 2011a). Additional samples of ice were collected with the same

method from the Icicle Room, as well as local exposures within the pile of breakdown at the entrance to the Frozen Freeway (Fig. 2). Samples of liquid water from pools along the edge of the Skating Rink were collected in 2018 and 2019 in 50-ml, screwtop vials with no head space.

Ice samples were melted overnight, and filtered to 0.2-μm the next day into new 15-ml tubes with no head space. These were

stored at 4°C before analysis (<48 hours later) of $\delta^{18}$O and $\delta$D in a Los Gatos Research DLT-100 water isotope analyser with a CTC Analytics autosampler at Brigham Young University. Analysis followed standard procedures (e.g. Perşoiu et al., 2011). Each sample was run eight times, along with a set of three standards, which were calibrated against VSMOW (Vienna Standard Mean Ocean Water). Precision of the resulting $\delta^{18}$O and $\delta$D measurements is ±0.2‰ and ±1.0‰, respectively. Stable isotope results were compared against values for the location of WWC downloaded from the Online Precipitation Isotopes Calculator

<waterisotopes.org>.

For glaciochemical analysis, remaining samples of meltwater were acidifed with trace element grade nitric acid and stored at 4°C before analysis with a Thermo Fisher iCAP Qc ICP-MS at Middlebury College following standard procedures (e.g. Kern et al.,



2011a). Each sample was run three times, with blanks and reference standards between every 5 samples. Samples were calibrated against a standard curve developed with 5 concentrations of an in-house standard based off NIST 1643f. Results were drift corrected, and detection limits were considered to be 1.0 ppb.

## 3 Results

### 3.1 Temperature dataloggers

The temperature data loggers ran continuously during their deployment, each recording 17,942 hourly measurements. When plotted as time series, these data reveal a repeating seasonal rhythm of temperatures within the cave, as well as air temperatures outside the cave entrance (Fig. 2). Temperatures at all three positions within WWC remained continuously <0°C. The coldest cave temperatures were recorded by Logger 3 in the Icicle Room (average of -1.5°C), which is closest to the cave entrance. There temperatures dropped to below -8°C at times each winter, although typically temperatures were closer to -4°C. The locations of Loggers 1 and 2 were warmer, -0.4 and -0.6°C, respectively (Fig. 3a). The records from these loggers also exhibit a great degree of similarity, and are strongly correlated with one another (r of 0.986, P=0.000). The temperature difference between these two loggers was lowest in the winter, and greatest in the summer.

The record from the logger outside WWC indicates a mean annual air temperature of 2.3°C (Fig. 3a). The winter of 2016–17 was colder than 2017–18 (based on the average temperature between November 1 and May 1). Both winters had a similar absolute coldest temperature of around -21°C. The winter 2017–18 had a longer stretch of time between the first and last subzero temperatures, but the beginning of this winter was relatively warm. As a result, the winter 2016–17 recorded a greater number of freezing degree days (950) in comparison with 766 during the winter of 2017–18. In response to this difference, the data loggers within the cave were approximately 0.5° C colder on May 1, 2017 relative to May 1, 2018.

Comparison of the records from all of the loggers indicates that when the outdoor temperature drops significantly below zero, the Icicle Room begins to cool, followed a few days later by the loggers deeper within the cave (Fig. 3b). In contrast, once the outdoor temperature stops cycling below zero at the beginning of the summer, then all loggers within the cave exhibit greatly reduced short-term variability, and temperatures at all of them rise asymptotically toward their equilibrium temperature, which is reached later in the summer (Fig. 2). Equilibrium temperatures are warmest at Logger 3 (Skating Rink, farthest back in the cave), and coldest at Logger 1 in the Icicle Room near the cave entrance (Fig. 2).

Close inspection of the time series also reveals interesting short-term behaviour at each of the logger sites. For instance, air temperatures in the Icicle Room (Logger 3) exhibit transient (~1 day) increases in air temperature during the summer on several occasions (Fig. 3). These are mirrored by simultaneous, but smaller amplitude, decreases in temperature at the loggers deeper within the cave (Fig. 3). Notably, after each of these disturbances, temperatures return to their baseline values. Another observation is that in May 2017, the temperature at Logger 2 rose above the temperature at Logger 1 for about 10 days, the only interval during the entire deployment when this reversal occurred (Fig. 3). Finally, in early June 2018, the temperature at Logger 3 abruptly rose about half a degree, and stayed elevated relative to the temperature at Logger 2 for the rest of the summer (Fig. 3).



### 3.2 Ground Penetrating Radar

GPR surveys along the Frozen Freeway, as well as in the Icicle Room, provided information about the thickness and internal stratigraphy of the ice. The survey in the Frozen Freeway extended for 75 m through passage of varying widths (Fig. 1c). Because the antenna was shielded, reflections from the cave walls and ceiling were not an issue. However, in the narrowest sections (less than 2 m wide), overlapping reflections from the cave walls beneath the ice surface made it difficult to identify consistent stratigraphy in the ice. Nonetheless, in wider sections, such as the Skating Rink, clear stratigraphic details within the ice are apparent in the radar data (Fig. 4). These take the form of sub-parallel banding at various depths, which are laterally consistent over several meters. There is a general trend of less pronounced reflectors in shallower parts of the ice, whereas the strongest reflectors are found at depths greater than a meter below the ice surface.

Along the entire length of the Frozen Freeway transect, a continuous reflector is discernible at depth, which likely represents the contact between the ice and the underlying bedrock (Fig. 4). Over the first 35 m of the transect, where reflections from the cave walls were not an issue, the maximum depth of this reflector (using a dielectric permittivity of 3.15) is between 2 and 2.5 m below the current ice surface.

The GPR transect in the Icicle Room spanned a length of 8 m. Results imaged layered ice superimposed on a sloping bedrock surface locally mantled by blocks of cave breakdown (Fig. 5). As in the Frozen Freeway, the upper 1 m of this ice exhibits only faint laminations, however below a depth of ~1 m the strength of the reflectors defining these laminations increases markedly. Based on the pattern of these parallel reflectors, and their relationship with the sub-adjacent, higher intensity reflectors from the bedrock, the maximum ice thickness in the Icicle Room, assuming a dielectric permittivity of 3.15, may exceed 3 m (Fig. 5).

### 3.3 Radiocarbon dating

Seven samples of packrat droppings collected from WWC were submitted for AMS radiocarbon dating. Six of these samples were successfully dated, yielding ages from 40 to 285 radiocarbon years, corresponding to median calibrated ages from AD 1890 to AD 1630 (Table 1, Fig. 6). The oldest of these samples has a 2-sigma calibration range that extends back to AD 1520. Four of the samples have 2-sigma calibration ranges that extend, with low probability, into the 20th century. One of these samples (WWC-FF-45) has a calibration range that overlaps (with very low probability) with post AD 1950 (Fig. 6).

The sample from the Icicle Room was collected from a vertical exposure of ice eroded by air currents entering from the Frozen Freeway. The depth of this sample was ~75 cm below the ice surface in August, 2016. Depths of samples from the Frozen Freeway were estimated visually relative to the height of the nearest ridge on the sublimated ice surface (Table 1). Overall, the estimated depths of these samples have no relationship with their ages. The oldest sample (WWC-FF-5) was from the shallowest estimated depth (5 cm). On the other hand, two samples obtained near one another from the same apparent depth (WW-20a and WW-20b) yielded identical ages.

### 3.4 Stable isotopes

A total of 94 samples collected from the continuous exposure at the back of the Skating Rink, and from isolated exposures of ice in and near the Icicle Room, were analysed successfully for $\delta^{18}O$ and $\delta D$. Overall, values of both isotopes are notably depleted, with an average $\delta^{18}O$ of -13.95 and $\delta D$ of -103.9 (Table 2). Minimum and maximum values of $\delta^{18}O$ are -17.56 and -8.66‰, and for $\delta D$ are -125.0 and -64.1‰. Overall, values of $\delta^{18}O$ and $\delta D$ are strongly and linearly correlated with one another ($r^2$ of 0.957).

... 





Given the wide range of isotope values in the samples from the Skating Rink exposure, a three-step screening was conducted to
identify outliers. First, values of d-excess in permil were calculated as d-excess = $\delta D-(8\times\delta^{18}O)$ (Dansgaard, 1964). Second,
because the distribution of the original data was not normal as revealed by a Shapiro-Wilk test (P=0.014), the values were log-
transformed. Third, values >2 standard deviations away from the mean were tagged as outliers. This process identified 3
samples, two with anomalously low values of d-excess, and one with an unusually high value (Fig. 7). These samples were
removed from further consideration, leaving 81 samples from this exposure.


On a plot of $\delta^{18}O$ versus $\delta D$, these 81 samples from the Skating Rink exposure plot parallel to and generally slightly below the
global meteoric waterline (Fig. 7). A linear regression through the 81 samples from the continuous exposure has a slope of 7.5
and y-intercept of 0.03 ($r^2$ of 0.980).

Plotting values of $\delta^{18}O$ and $\delta D$ vs. depth in the 2-m high exposure reveals short-term variability superimposed over a longer
trend (Fig. 8). In general, isotopic values in deeper samples are more depleted, and rise toward the top. The most enriched
samples were collected from within a few centimetres of the cave ceiling. The overall trend in d-excess runs generally contrary
to $\delta^{18}O$ and $\delta D$, with higher values in deeper ice.

In the field, the upper 1.5 m of this exposure, which was more accessible, was visibly stratified, with thin layers of fine carbonate
precipitates defining boundaries between 13 discernible layers. Similar layers were not noticed in the bottom 50 cm of the
exposure, however this may simply reflect the fact that it was difficult to view the bottom of the exposure head-on. Some of
these layers, which range from 2 to more than 20 cm thick, correspond with variations in $\delta^{18}O$ and $\delta D$ (Fig. 8). For instance, $\delta$
values are notably enriched in Layer 2, with $\delta^{18}O$ up to -8.7‰. Similarly, Layer 8 and the central part of Layer 9 both contain
relatively enriched ice. However, isotopic values do not always change appreciably across these layer boundaries. For example,
$\delta^{18}O$ and $\delta D$ rise steadily through Layer 5, continuing a trend that spans from Layers 4 through 6 (Fig. 8). The boundary
between Layers 10 and 11 appears is the only instance in which a visible boundary coincides with a major change in isotope
values (a shift in $\delta^{18}O$ of 1.2‰ and $\delta D$ of 11.5‰).

### 3.5 Glaciochemistry

A total of 13 major and trace elements were detectable in ice samples analysed from the exposure at the Skating Rink. Ranked in
order from most to least abundant they are Ca, Al, Mg, K, Na, P, Si, Ni, Ti, Ba, Sr, Mn, and Rb (Fig. 9). Peak abundances of
calcium are ~20,000 ppb, and Al, Mg, K, and Na are all detectable in some samples at concentrations above 1000 ppb.
Rubidium is the least abundant detectable element, generally <1 ppb. Phosphorous, Si, and Sr were not detectable in some
samples. A principal component analysis applying a varimax rotation to log-transformed values of Ca, Al, Mg, K, Na, Ti, Ba,
Mn, and Rb (removing P, Si, and Sr which were not detectable in all samples) loads all elements except Al and Ni on the first
principle component (PC-1) at values from 0.763 to 0.890 (Fig. 8). These elements follow a consistent pattern of generally low
values in the bottom half of the exposure, rising values between ~100 and 50 cm depth, and notably elevated values in the
uppermost 50 cm (Fig. 8). In contrast, Ni exhibits the strongest positive correlation with PC-2 (0.891), and Al is strongly and
negatively correlated with PC-3 (-0.827). Nickel has general low and stable values until the uppermost sample, whereas Al
abundance is high between 125 and 150 cm, and again around a depth of 50 cm (Fig. 10).



Similar elemental abundances were measured in the ice samples collected between the Frozen Freeway and the Icicle Room. Calcium is again the most abundant element, with an average concentration of 4850 ppm, and Rb is the least abundant (<1 ppm). Mean abundances for all elements in these samples are strongly correlated with those measured for the Skating Rink exposure ($r^2$

of 0.945). Unlike in the exposure at the Skating Rink, there is no obvious trend in abundance in these samples. However, because they were collected from isolated locations where ice was locally visible between blocks of breakdown, their relationship with one another is unclear and it may be inappropriate to consider them parts of a stratigraphic sequence.

## 4 Discussion

### 4.1 Origin, extent, and age of the ice

The data loggers that recorded air temperatures between 2016 and 2018 provide a clear explanation for why perennial subzero conditions are maintained inside WWC despite a mean annual temperature of 2.3°C outside the entrance. As illustrated in Figure 2, cold air enters the Icicle Room (Logger 3) anytime the outside air temperature drops notably below ~0°C. If cold temperatures outside are maintained for more than a few days, this cold pulse penetrates deep enough into the cave to be recorded by Logger 2 in the Frozen Freeway and Logger 1 at the Skating Rink (Figure 3b). This pattern indicates that a

connection exists between the currently accessible end of WWC and the surface of the plateau ~100 m above the cave. This conduit acts as a chimney, allowing relatively warm air within the cave to rise towards the surface in the winter (Balch, 1900; Luetscher and Jeannin, 2004b; Thury, 1861). This air is replaced by cold air flowing in through the cave entrance into the Icicle Room, refrigerating the cave interior. In contrast, during the summer, there is a minor reversal of this pattern as cold air within the cave flows out through the entrance, replaced by warm air penetrating downward through the hypothesized connection

between the cave and the plateau surface. However, because the entrance of the cave is ~15 m higher than the Frozen Freeway (Fig. 1c), it is not possible for all of the dense, cold air to be evacuated from the cave in the summer. As a result, the interior of WWC is disproportionately impacted by cold air in winter, and is relatively unaffected by summer warmth.

Occasionally during the summer months the temperature in the Icicle Room abruptly jumps ~1°C before returning to baseline

levels (Fig. 3c & 3d). This shift is accompanied by a minor temperature decrease at the loggers deeper in the cave. This pattern indicates that some process brings relatively warm outside air into the Icicle Room, displacing colder air deeper into the cave, but not for long enough to have a lasting impact on the overall temperature distribution. Comparison of the logger records with meteorological data from the Chepeta RAWS (remote automated weather station) 60 km to the east (Fig. 1b) at a slightly higher elevation (3680 m) indicates that each of these transient temperature increases was associated with wind from 320 to 340° at

velocities greater than 15 m/s. Given the orientation of the cave entrance, this azimuth is perfectly aligned to push relatively warm outside air through the cave entrance, temporarily warming the Icicle Room and displacing cold air deeper into the cave. However, because the general airflow is outward during the summer, as soon as the wind direction or velocity changes and warm air is no longer forced into the cave, the pattern reverses and the Icicle Room cools down again.

Given the distribution of ice deposits, it appears that the water responsible for the ice in WWC entered from the back, flowing through the boulder-choked constrictions, as well as through the entrance into the Icicle Room. Because there are no streams or lakes on the landscape above the cave, this water is likely derived from infiltrating precipitation. Furthermore, given the climate at this location, much of this water is likely related to the melting of winter snowpack in late spring. This rapid melt produces a pulse of water that penetrates downward through the epikarst to reach the back of the cave, perhaps following the conduit that





allows cave air to rise toward the surface in the winter.  Snowmelt on the north-facing cliff above the cave entrance can enter the Icicle Room directly.  It is also possible that intense summer rainstorms may deliver water to the plateau above WWC in quantities sufficient to flood the epikarst and deliver water to the cave interior.  Either way, this water freezes upon entering the subzero cave interior, incrementally adding a new layer to the perennial ice deposit.  Given this model for maintenance of freezing temperatures and the formation of ice from inflowing water, WWC is classified as a "dynamic cave with congelation

ice" (Luetscher and Jeannin, 2004a).

Winter Wonderland Cave was not visited during the summer of 2017, so it is unknown whether water entered the cave that year. However, in August 2018, liquid water and new ice were noticed at the Skating Rink and along parts of the Frozen Freeway. The temperature data logger from the Skating Rink (Logger 1) recorded a ~1°C increase in temperature during the first week of

June, 2018 (Fig. 3d).  In contrast to the summer temperature increases in the Icicle Room, this temperature rise was long-lasting: during the weeks leading up to this point the temperatures at Logger 1 and Logger 2 were identical, yet after this point, Logger 1 at the Skating Rink remained ~1 °C warmer than Logger 2.  The arrival of relatively warm meltwater to the back of the cave at this time could explain the abruptness of this temperature jump, and the slow release of latent heat as this water froze could have maintained the slightly warmer temperatures relative to Logger 2.  Snowpack monitoring sites at Trial Lake and Brown Duck, at

similar locations, elsewhere in the southwestern Uinta Mountains rapidly lost the majority of their snow water equivalent in May, 2018 (Fig. 1b).  This offset suggests that it requires ~1-2 weeks for meltwater to transit from the plateau surface to the cave.

Results from the GPR surveys indicate that the ice forming the floor of the Frozen Freeway is generally from 1 to ~2.5 m thick and has accumulated over an uneven surface of bedrock and blocks of breakdown (Fig. 4).  The 2-m high exposure of ice at the

edge of the Skating Rink starts ~45 cm above the floor of the Frozen Freeway, exploiting remnant ice layers reaching to the cave ceiling that have not been removed by sublimation.  Given the assumed dielectric permittivity, therefore, it is possible that as much as a meter of additional ice exists below the deepest stratigraphic level accessible in this exposure.  Similarly, in the Icicle Room, an ice thickness possibly in excess of 3 m was imaged (Fig. 5).  In both locations, the tendency of the upper ~1 m of ice to exhibit reduced stratigraphic layering compared with the deeper ice may reflect a greater presence of mineral precipitates, dust,

or organic matter concentrated specific levels within the deeper ice.

Determining the age of cave ice deposit is rarely straightforward, and the situation in WWC is no different.  Compared with a sag-type cave in which snow and organic matter accumulate in a vertical shaft (e.g. Munroe et al., 2018; Spötl et al., 2014), the interior of WWC is relatively devoid of organic matter.  On the other hand, the radiocarbon dated packrat droppings do provide

important age constraints (Fig. 6).  Because these samples were collected from the uppermost layers of the ice deposit, they provide minimum limits on the age of the ice, suggesting that most of the ice accumulated before ~AD 1850.

## 4.2 Paleoclimate implications

In vertical exposures, the ice in WWC is composed of visually distinct layers from <5 to ~20 cm thick delineated by mineral precipitates and changes in bubble density.  Layering on a similar scale is also resolvable in the GPR results (Figs. 4 & 5).  If

these layers were constructed through the incremental additions of thin films of water, then the mineral precipitates defining the visible layer boundaries could have formed in response to sublimation that lowered the ice surface creating a lag of material that was originally included in the ice.  Alternatively, the visible layers may have formed through slow, closed-system freezing of pools of water with a depth equivalent to the resulting layer thickness.  Mineral precipitates would have formed within each pool





as the floating ice thickened downward, concentrating the residual liquid water. These two models have been described as "floor
ice" and "lake ice" respectively in previous ice cave research (Perşoiu et al., 2011).

Downward freezing of a layer of water beneath a thickening lid of ice will result in sequentially more depleted ice at depth, as
the heavier [18]O and deuterium are preferentially incorporated into the early-formed ice (Citterio et al., 2004b; Persoiu and
Pazdur, 2011; Souchez and Jouzel, 1984). Thus, plotting $\delta^{18}O$ and $\delta D$ vs. depth, and highlighting the locations of visually
identified layer boundaries, was employed to determine the likely origin of the different ice layers. When plotted this way (Fig.
8), several layers exhibit trends of increasing isotope depletion with depth, however these differences are generally <1‰ for $\delta^{18}O$
from top to bottom in a given layer. The exception is Layer 8 where $\delta^{18}O$ falls from -12‰ to -15‰ over a layer thickness of 14
cm. Previous work has shown that a top to bottom offset in $\delta^{18}O$ of ~4‰ occurs within 5 to 10-cm-thick layers of lake ice in a
well-studied Romanian ice cave (Perşoiu et al., 2011; Persoiu and Pazdur, 2011). With this observation as a reference point,
Layer 8 in WWC likely formed through closed-system freezing of a thick layer of water, but the smaller $\delta^{18}O$ offset in the other
layers is not necessarily a signal of top-down freezing that shifted isotopic values from the composition of the original water.

Another way to evaluate likelihood that ice formed through close or open-system freezing is to plot $\delta^{18}O$ vs. $\delta D$ from an
individual layer and consider whether the values align with the (local) meteoric water line, indicating floor ice, or with a lower
"freezing slope" indicative of fractionation, typical of lake ice (Jouzel and Souchez, 1982; Perşoiu et al., 2011). When arranged
this way, sets of samples (n≥3) defining individual layers in the Skating Rink exposure have slopes from 6.0 to 13.7 (Table 2).
Layers 6, 9, 11, 12, 13, and samples from the bottom of the exposure have slopes that are close to the meteoric water line. The
isotopic measurements in these layers, therefore, likely record of the composition of the original meltwater with fidelity. In
contrast, Layer 7 has a slope <6.0, which is consistent with fractionation during freezing. The slopes of Layers 4 and 10 are
quite high (>12), although they were determined for just three points. Previous experimental and field-based work has
demonstrated that isotopic composition of the original water can be found at the intersection of a line fit to the $\delta^{18}O$ and $\delta D$
measurements for an individual ice layer and the meteoric waterline (Jouzel and Souchez, 1982; Perşoiu et al., 2011). Applying
this approach yields estimates of original $\delta^{18}O$ and $\delta D$ values for the layers that appear to have fractionated during closed-system
freezing (Table 2). Two cautions must be noted in considering this analysis. First, samples were not collected continuously
through the exposure, they were spaced 2 cm (for the upper part) and 5 cm (for the lower part) from one another. As a result, not
all ice was sampled and the isotopic composition of the ultimate top and bottom ice in a given layer is unknown. Second,
previous work has noted that kinetic effects during rapid freezing can cause deviations from the freezing slope (Perşoiu et al.,
2011). Based on relationships between $\delta D$ and d-excess, kinetic effects likely played a role in the formation of at least some
layers in the Skating Rink exposure. Nonetheless, with these considerations in mind, estimated isotopic compositions of the
original water obtained from the intersection of the freezing slope and meteoric water line were combined with the average
isotopic values for layers apparently did not fractionate, based on downward trends in isotope values and $\delta^{18}O$ vs. $\delta D$ slopes.
This approach yielded a composite series of $\delta^{18}O$ and $\delta D$ values vs. depth (Fig. 8).

The overall pattern in these data is that the deep ice is more depleted in heavy isotopes, whereas stratigraphically higher ice is
more enriched (Fig. 8). This trend indicates a long-term change in the isotopic composition of the water entering WWC. Air
temperature exerts a well-documented effect on isotopic values of precipitation, with a lower temperature of condensation
corresponding to more depleted $\delta^{18}O$ and $\delta D$ (Dansgaard, 1964). Therefore, one possible interpretation is that average air
temperatures increased over the time period represented by the Skating Rink exposure. However, given the relationship between





monthly average temperature and estimated δ¹⁸O for precipitation at the Brown Duck SNOTEL site (Fig. 1b), the difference
between the most depleted (-16.2‰) and most enriched (-8‰) samples from this exposure would correspond to a temperature
change of approximately 11°C. This result exceeds the temperature difference between full glacial and modern conditions in the
Uinta and Wasatch Mountains determined from numerical modelling of paleoglaciers (Laabs et al., 2006; Quirk et al., 2018,
2020). Unless the ice in WWC is residual from the last glaciation, which reached its maximum in this area at 18-20 ka (Laabs et
al., 2009; Quirk et al., 2020), it is unlikely that changes in the isotopic values from WWC are solely a function of temperature.

A more plausible interpretation is that the isotopic variability reflects changes in the relative abundance of seasonal precipitation
over WWC. Under the modern climate, the Uinta Mountains are influenced by two distinct precipitation patterns strongly
limited to specific seasons. Storms penetrating inland from the Pacific Ocean produce a peak of precipitation during December,
January, and February (Mitchell, 1976). By virtue of the cold winter air temperatures, and the long transport pathway for this
moisture passing over multiple mountain ranges, this precipitation is strongly depleted in ¹⁸O and deuterium. In contrast, during
the summer, primarily in August, moisture with relatively enriched isotopic values is delivered from the south by the North
American Monsoon circulation (Metcalfe et al., 2015; Ropelewski et al., 2005). Correspondingly, the Online Isotopes in
Precipitation Calculator predicts that the location of Winter Wonderland Cave receives precipitation with δ¹⁸O around -23‰ in
winter, and around -8‰ in August.

The rapid 1°C increase in air temperature recorded by the logger at the Skating Rink in early June, 2018 is a likely signal for the
arrival of snowmelt at the beginning of that summer (Fig. 3d). However, liquid water was still observed at the Skating Rink
more than two months later on August 29, 2018. Similarly, although there is no temperature record for the summer of 2019,
liquid water was again present at the Skating Rink on August 19, 2019. Assuming that the snowmelt pulse usually arrives at the
beginning of the summer, matching the annual timing of snowmelt, the persistence of a relatively thin layer (centimeters) of
liquid water in subzero conditions for multiple months is improbable. Therefore it is likely that additional water entered the cave
later in the summers of 2018 and 2019, after the main snowmelt pulse. This summer water would most conceivably reflect the
monsoonal precipitation that dominates in August. In this model, therefore, the water forming ice in WWC represents an
integrated snowpack signal that is augmented by some amount of isotopically heavier summer precipitation.

The isotopic composition of the snowmelt directly above WWC is unknown. However, fully integrated snowpack samples
collected at the elevation of the plateau above the cave at Pole Mountain ~50 km eastward along the south flank of the Uinta
Mountains in late March, 2019 had an average δ¹⁸O of -18.9‰ (Fig. 1b). Similar full-snowpack samples collected at the same
elevation in early April, 2019 at Trial Lake ~25 km northwest of WWC had an average δ¹⁸O of -20.3‰ (Fig. 1b). The mean of
all these values (-19.6‰) is lower than the value of -18.2‰ for two samples of water collected from the Skating Rink surface in
August, 2019, indicating that the water collected in the cave that summer was a mixture of depleted winter snow and enriched
summer rain.

With this insight, a mixing model was constructed to predict the δ¹⁸O value of water comprised of varying proportions of
snowmelt (at -19.6‰) and monsoonal rain (-8‰). This model suggests that the water collected from pools on the surface of the
Skating Rink in late summer, 2019 was a mixture of 88% winter snowmelt and 12% monsoonal rain. This estimate of the
summer component in this sample is a minimum, because the water had already begun to freeze beneath a lid of ice, which
would drive δ¹⁸O in a more negative direction (Jouzel and Souchez, 1982; Perşoiu et al., 2011). Nonetheless, the value of δ¹⁸O



in the modern water is more negative than any of the analysed ice samples, indicating that either this water contained a lower

fraction of summer rain than the ice samples, or that the snowmelt pulse in 2019 was unusually large.

Going one step farther, this model can be applied to the average (measured and reconstructed) values of δ¹⁸O for ice layers in the Skating Rink exposure (Fig. 8). This approach suggests that the deepest, oldest ice formed from water containing ~40% summer precipitation, and that the fraction of monsoonal moisture incorporated in this ice increased to ~60% near the top of the exposure.

Two outliers to this pattern are notable: a peak value of 83% in Layer 5, and a lower value of 38% in Layer 7. However, Layer 5 was thin and is represented by just 3 datapoints, creating uncertainty in its regression slope that carries over into the estimate of original water composition. Layer 7 contains two sets of isotopic values, a constant upper half and a lower half with a pronounced negative excursion (Fig. 8), implying that the suite of samples collected may have inadvertently spanned two layers separated by a contact that went unnoticed in the field. An obvious limitation is that this simple model assumes that the isotopic

composition of snowmelt and monsoonal precipitation were constant over time. It also assumes that the interpolated isotopic value for August precipitation from the OPIC is accurate, and that the snowpack samples adequately represent the composition of the snow on the plateau above WWC. Nonetheless, even with these considerations, it is notable that the relative importance of winter and summer precipitation may have changed at this location over time.

The radiocarbon dated organic remains were collected within the uppermost layers of ice forming the floor of the Frozen Freeway, which corresponds to Layers 7 and 8 in the Skating Rink exposure (Fig. 8). This relationship allows the exposure to be divided broadly into a lower part, where the ice apparently accumulated prior to the 1800s, and an upper part (Layers 1 through 6), which accumulated more recently. The lower layers, therefore, correspond with the interval of generally cooler temperatures in the northern hemisphere referred to as the Little Ice Age (Grove, 1988). The reconstructed summer water content of this ice

varied in a narrow range between 40 and 50%, suggesting relatively consistent conditions under which the snowmelt pulse and summer monsoon contributed roughly equal amounts of water to the cave. After this time, summer water began to exert a greater control over the isotopic composition of the ice.

Records of δ¹⁸O from lake sediments on the White River Plateau in northwestern Colorado (Fig. 1a) suggests that winters during

the Little Ice Age were notably snowy, with snowfall making up the greatest fraction of annual precipitation at any time during the Holocene (Anderson, 2012). After that time, snowfall became less important. A similar pattern is seen in multiproxy lacustrine records from San Luis Lake (Fig. 1a) in southern Colorado (Yuan et al., 2013). The suggestion that water derived from the melting of winter snow contributed to a greater fraction of the ice forming in WWC at this time is consistent with these other regional hydroclimate records. Values of δ¹⁸O from an exposure of ice in Strickler Cavern (Fig. 1a) record a shift from -

14.5 to nearly -17‰ over the first few decades of the 18th Century, which was interpreted as an interval of decreasing temperatures leading to the peak of the Little Ice Age (Munroe et al., 2018). An additional point of reference is provided by isotope measurements in a glacier ice core from the Wind River Mountains (Fig. 1a), ~400 km to the northeast of WWC (Naftz et al., 1996). In this core, a positive shift in δ¹⁸O of ~1‰ accompanied the end of the Little Ice Age around AD 1845 (Schuster et al., 2000). The magnitude of this shift is about half as large as the change in δ¹⁸O between the bottom of Layer 6 and the top

of the Skating Rink exposure (excluding the unusually enriched samples from Layer 2). Part of the trend toward more enriched δ¹⁸O values in the uppermost part of the Skating Rink exposure may, therefore, reflect an increase in average air temperature after the Little Ice Age ended.



### 4.3 Glaciochemistry interpretation

The elemental abundances measured in the ice from WWC are notably similar from sample to sample, suggesting that the composition of the water did not changed dramatically over time. Such constancy fits the assumption that the majority of the dissolved load in the water was obtained through water-rock dissolution reactions in the epikarst. If the pathways leading water from the plateau surface down to the cave did not change appreciably over the time period represented by the ice, then it is logical that the overall chemical signature of this ice was relatively constant.


At the same time, however, the majority of elements exhibit a notable increase in concentration in the upper part of the ice. This pattern is consistent across nearly every detectable element, and is clearly seen in the first principle component (PC-1) calculated from these elemental abundances (Fig. 8). The shift toward greater elemental enrichment occurred primarily as a step change in Layer 8, which as noted above contains organic remains dating to the latter part of the Little Ice Age. Thus it seems that the

glaciochemical composition of this ice changed between the Little Ice Age and the years that followed. Consideration of the setting of WWC and available information about regional hydroclimate in the latest Holocene reveals possible explanations for the rising elemental abundances in the younger ice. First, the decrease in winter snowpack noted in the lacustrine records (Fig. 1a) from Colorado (Anderson, 2012; Yuan et al., 2013) could have slowed the rate of infiltration through the epikarst, affording water a longer contact time with host rock during transit from the ground surface to the cave, leading to an increased dissolved

load in the water. There is a slight tendency toward higher molar Mg/Ca ratios in the upper part of the Skating Rink exposure (Fig. 8), consistent with more evaporative conditions in the epikarst (Steponaitis et al., 2015). Furthermore, if a greater fraction of the water reaching the cave was coming from summer monsoonal moisture, rather than winter snowmelt, near-surface rock temperatures in the epikarst may have been elevated above Little Ice Age values, increasing rates of dissolution reactions. Either way, the increase in dissolved load in the ice roughly at the same time as the shift toward an apparently higher fraction of

summer moisture, along with an overall decrease in the observed thickness of the ice layers, is consistent with a shift toward reduced winter snowfall.

One element that notably does not follow the pattern of increasing abundance in the uppermost ice is Al, which is most abundant in samples from Layer 12 and Layer 7 (Fig. 8). Aluminium is not present in significant quantities in the carbonate bedrock

hosting WWC, suggesting that the Al in the ice has another source. Previous work has documented the abundance of Al in Uinta Mountain dust, where Al in the form of aluminosilicates is present at a weight percent abundance of ~7% (Munroe, 2014). Thus the changing abundance of Al in the Skating Rink exposure may reflect variations in the abundance of dust brought into the cave by inflowing air currents during winter. This raises the possibility that additional age control, combined with the collection of more ice samples, could allow reconstruction of dust abundance over time from this layered ice deposit.


A final observation in the glaciochemistry is the elevated level of Ni in the uppermost samples. After averaging 16.5 ppb through nearly all of the Skating Rink exposure, Ni rises in the uppermost 10 cm, and reaches a maximum of 500 ppb in the youngest sample immediately below the cave ceiling (Fig. 8) Work elsewhere in the Uinta Mountains has demonstrated that the abundance of Ni is greatly elevated in modern dust (Munroe, 2014), as well as in lake sediments that accumulated after AD 1900

(Reynolds et al., 2010). This dramatic elevation of Ni abundance has been ascribed to fugitive dust from mine tailings and other anthropogenic activities in the southwestern US. In WWC, the increase in Ni levels begins to rise in ice approximately 30 cm stratigraphically higher than the radiocarbon ages (Fig. 8), a relationship that adds further support for the theory that the uppermost ice accumulated after the Little Ice Age.



### 4.4 Indications of recent change

Studies of other ice caves have reached the sobering conclusion that many subterranean ice bodies have a negative mass balance and are rapidly disappearing as a direct or lagged response to climate change above the cave (Kern and Perşoiu, 2013; Pflitsch et al., 2016). Winter Wonderland Cave was discovered less than 10 years ago and has only been entered five times, thus the period and number of observations is minimal. However, these repeat visits have identified notable evidence suggesting that conditions within the cave have recently changed. During summer visits in 2014, 2015, and 2016 the surface of the ice along the Frozen

Freeway was dramatically sculpted into a series of ridges and furrows presumably created by sublimation induced by air currents (Fig. 10a & 10b). The surface of this ice was mantled by cryogenic mineral precipitates, and no liquid water was observed anywhere in the cave. The overall impression was that it had been many years since water entered the cave and added a new layer to the ice deposit. Short-term observations of an ice cave in the Canadian Rockies suggests that sublimation producing sculpted patterns on the surface of ice proceeds at a rate on the order of 3 mm per year (Marshall and Brown, 1974). Thus, the

~30 cm of relief on the surface of the Frozen Freeway suggests that liquid water had not entered the cave for possibly as long as 100 years. As noted earlier, that situation changed dramatically in 2018 and again in 2019 when considerable water entered from the rear of the cave and flooded across the Frozen Freeway, filling most of the troughs and submerging most of the crests (Fig. 10c & 10d). Certainly, infiltration of liquid water is not a new occurrence in WWC; it is exactly this process which created the layered ice deposit in the first place. On the other hand, the abrupt appearance of this water in 2018 following an apparently long

stretch of time in which water failed to reach the cave, suggests a significant shift in the availability of water, or the thermal environment within the cave. Neither the water years 2017-18 or 2018-19 featured unusual amounts of precipitation, but temperatures at the Brown Duck snowpack monitoring site (Fig. 1b) have risen at a rate of 0.8°C/decade over the past 25 years. Thus, it is possible that warming temperatures have crossed a threshold and begun to melt ice farther back in an inaccessible part of the cave, contributing to the meltwater observed in the Skating Rink in the last two summers. Melting of ice with more

negative isotope values could also explain why the water collected in 2019 was so depleted in $\delta^{18}O$ and $\delta D$.

### 4.5 Limitations and direction for future research

This multidisciplinary investigation of the perennial ice in Winter Wonderland Cave supports a model for the origin of the ice, provides constraints on the age of the ice, and uses isotopic and glaciochemical data to explore the possible paleoclimate

significance of this ice. Despite these achievements, this work remains a preliminary assessment, and there are a number of significant directions in which future efforts could be focused. Foremost among these would be the acquisition of additional age control. The collection and radiocarbon dating of organic remains from other locations within the cave would be helpful in further limiting the age of the ice. Similarly, although uranium abundance was not assessed in the glaciochemistry reported here, it may be possible that the ice could be directly dated with U/Th techniques (Cheng et al., 2013).


In terms of sampling, it may be worthwhile to attempt to retrieve a core where the GPR analysis indicates a maximum ice thickness. Given the difficult access to the cave and a challenging working environment, retrieval of such a core would not be a simple process. However, an ice core would allow the analysis of truly contiguous samples from an unambiguous stratigraphic context. Additional environmental proxies in the ice, for instance pollen (Feurdean et al., 2011), black carbon, or dust could be

studied in a core to reveal more details about environmental conditions outside of the cave when the ice was accumulating.



Either combined with coring, or through a separate sampling campaign, it would also be useful to retrieve intact, oriented blocks of ice for crystallographic study (Holmlund et al., 2005). Analysis of crystal orientations, bubble trails, inclusions, and other features would provide information about the conditions under which the ice formed (Citterio et al., 2004a; Marshall and Brown, 540   1974). Such evidence could be useful in distinguishing between open and closed-system models for the genesis of the ice layers.

Additional data loggers could be deployed throughout the cave to further collect micrometeorological data. Relative humidity loggers were installed in the cave in 2019, and should be useful in identifying changes in the direction of cave ventilation. Such efforts could be combined with deployment of a three-dimensional, ultrasonic anemometer (Pflitsch and Piasecki, 2003). More 545   data loggers from the rear of the cave might be able to better identify the thermal pulse associated with the arrival of meltwater in early summer. Similarly, deployment of a thermal camera could further document the arrival of this water (Berenguer-Sempere et al., 2014).

Finally, in terms of more traditional cave research techniques, a focused campaign to collect summer precipitation and winter 550   snow on the plateau above WWC would constrain the initial properties of the infiltrating water and permit more nuanced interpretations of the source for the ice and its dissolved load (Carey et al., 2019; Herman, 2019). Similarly, speleothems are present in the cave. None of them are currently active and no dripwater was observed during any of the August visits. However, it would be interesting to determine when these speleothems were growing and what they may contain as a paleoclimate record.

## 5 Conclusions

Winter Wonderland Cave is a solution cave in Carboniferous limestone at an elevation of approximately 3140 m in the Uinta Mountains of northeastern Utah, USA. The cave contains a layered perennial ice deposit formed by freezing of infiltrating water. Radiocarbon dating of organic remains indicates at least some of this ice accumulated before approximately AD 1850. Ground penetrating radar indicates that the ice is locally ~3 m thick. Isotopic analysis suggests that some ice layers may have formed during closed-system freezing, however most samples plot near the meteoric water line. The ice seems to have been 560   formed from a mixture of winter snowmelt and summer precipitation, with winter water dominating in the early part of the record, and summer water becoming more significant in the latter part. This shift is consistent with other regional hydroclimate evidence suggesting that winters were unusually snowy during the Little Ice Age. Glaciochemical data indicate that the overall dissolved load in the ice increased after the Little Ice Age. This change may be a sign of reduced rate of infiltration that increased water-rock interaction time. Abundances of Al and Ni follow a different pattern, and likely reflect variation in the 565   amount of eolian dust incorporated in the ice over time. The appearance of liquid water in the cave in the last two summers is unusual, and may be a sign of a recent change in the cave environment. Collectively these results emphasize the potential significance of the ice in Winter Wonderland Cave as a paleoclimate record, and highlight its vulnerability. Future work should focus on acquiring additional age control for this ice and possibly retrieving a continuous core through the ice deposit.

*Acknowledgments.* David Herron (USDA Forest Service) discovered Winter Wonderland Cave and provided the detailed mapping from which Figure 1 was simplified. Middlebury students Quinn Brencher, Kristin Kimble, Miranda Seixas, and Caleb Walcott were instrumental in the success of this challenging fieldwork. Thanks to Greg Carling and Kevin Rey at Brigham Young University for assistance with snow collection and the stable isotope measurements, and to Pete Ryan and Jody Smith at Middlebury College for help with the ICP-MS analysis.



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



**Figure 1. Location map of the study area. (a) Western North America with US state outlines. The state of Utah is highlighted. Red box**
**denotes the location of the enlargement in panel (b). Other sites noted in the text are identified: SC-Strickler Cavern (Munroe et al., 2018), WRR-Wind River Range (Naftz et al., 1996), WRP-White River Plateau (Anderson, 2012), SLL-San Luis Lake (Yuan et al., 2013). (b) Enlargement of the Uinta Mountain region in USA NAIP natural color imagery. The yellow triangle marks the location of Winter Wonderland Cave (WWC). Other sites noted in the text are identified: TL-Trial Lake snowpack monitoring site and snow sampling site, BD-Brown Duck snowpack monitoring site, PM-Pole Mountain snow sampling site, CR-Chepeta RAWS. (c) Profile and**
**map views of Winter Wonderland Cave simplified from mapping by David Herron, USDA-Forest Service. Cross-hatching pattern denotes breakdown and blue color denotes perennial ice. The locations of the cave entrance, the Icicle Room, the Frozen Freeway, and the Skating Rink are noted. The orange lines mark the Ground Penetrating Radar transects. The red star at the end of the Skating Rink is the exposure described in the text. The boxed numerals 1-3 identify the locations of the temperature dataloggers.**



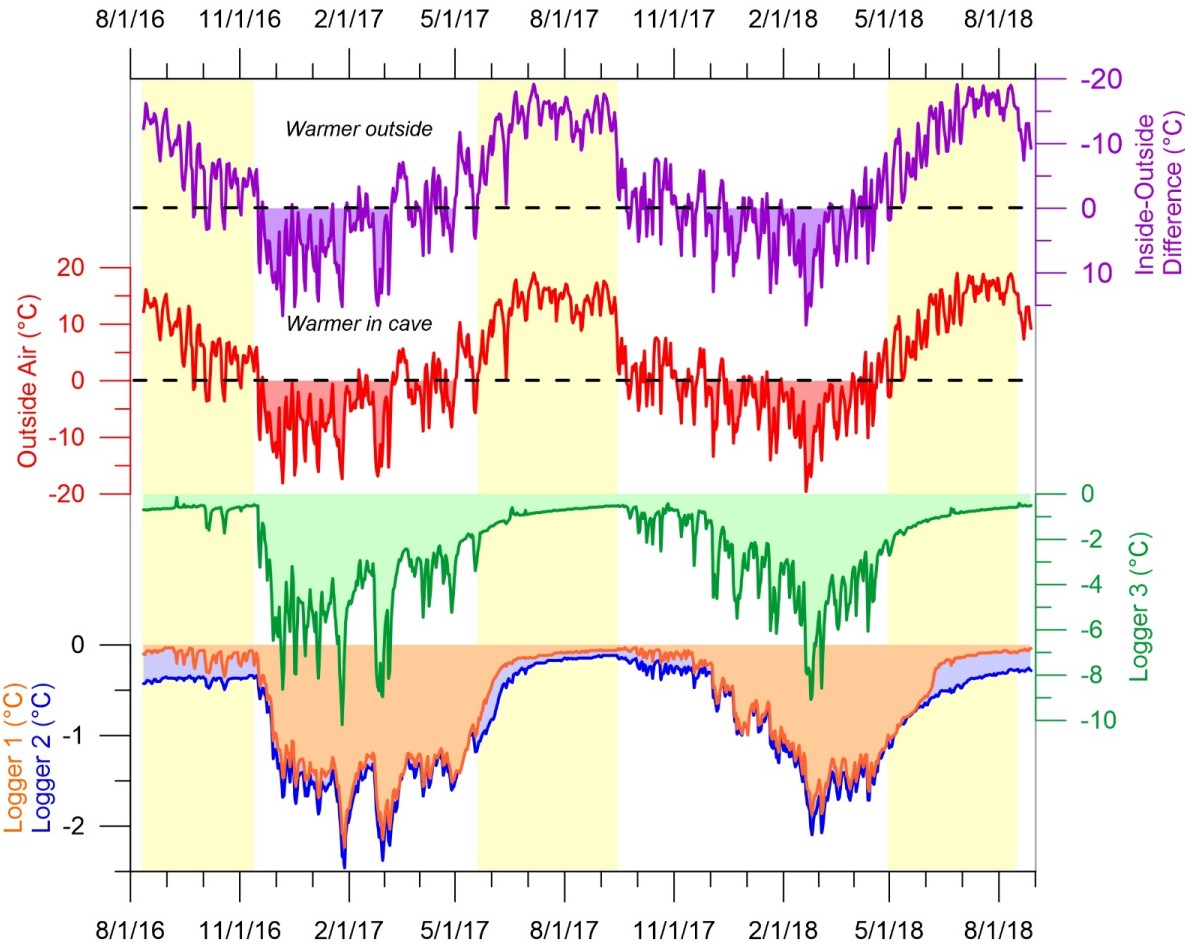

**Figure 2. Temperature records from dataloggers deployed in Winter Wonderland Cave from August 2016 through August 2018. Logger 1 was at the Skating Rink, Logger 2 along the Frozen Freeway, and Logger 3 in the Icicle Room (Fig. 1c). Outside air temperature was recorded by a logger in a solar radiation shield outside the cave entrance. Yellow shading highlights the summer periods when cave ventilation is greatly reduced. The purple time series at the top displays the difference between the temperature at Logger 1 and the temperature outside the cave. Time series from loggers within the cave are filled below 0° C. Dashed black lines**
**highlight 0° C on the outside air and the temperature difference plots.**





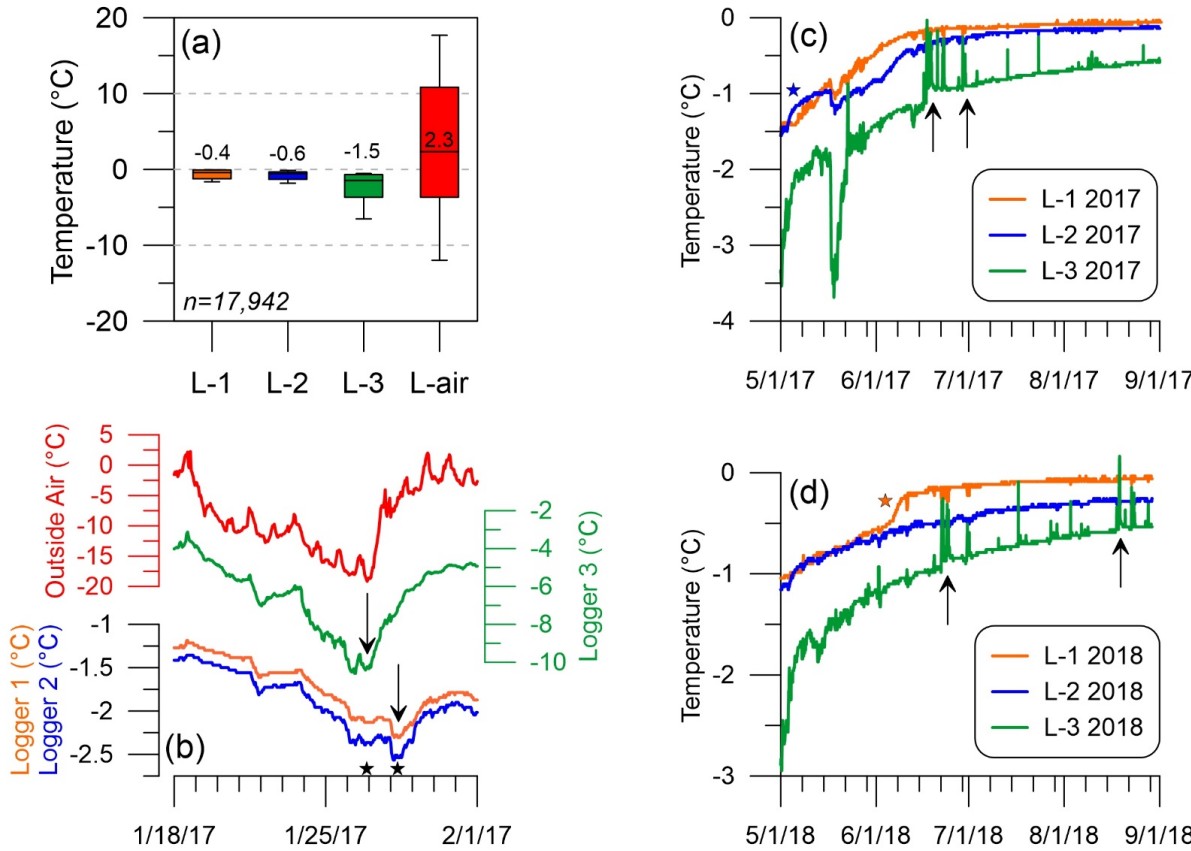

**Figure 3. Details of temperature records from Winter Wonderland Cave.** (a) Box and whisker plot showing the distribution of temperature measurements at each of the three loggers (L-1 through L-3), as well as outside the cave. (b) detailed view of temperature records from late January, 2017. A notable low point in temperature outside the cave was mirrored immediately by a temperature drop within the Icicle Room, recorded by Logger 3 on January 27 (star). A corresponding drop in the temperature deeper in the cave was recorded two days later, supporting the theory that cold air enters through the Icicle Room in winter. (c and d) Temperatures during the summer of 2017 (c) in the summer of 2018 (d) at the loggers within the cave. The black arrows highlight examples of transient warming in the Icicle Room that were not observed deeper in the cave. The blue star in (c) marks an interval in early May when the temperature at Logger 2 in the Frozen Freeway briefly rose above the temperature of Logger 1 at the Skating Rink. The orange star (d) highlights a dramatic, permanent increase in temperature at Logger 1, interpreted to represent the arrival of meltwater at the back of the cave in 2018.



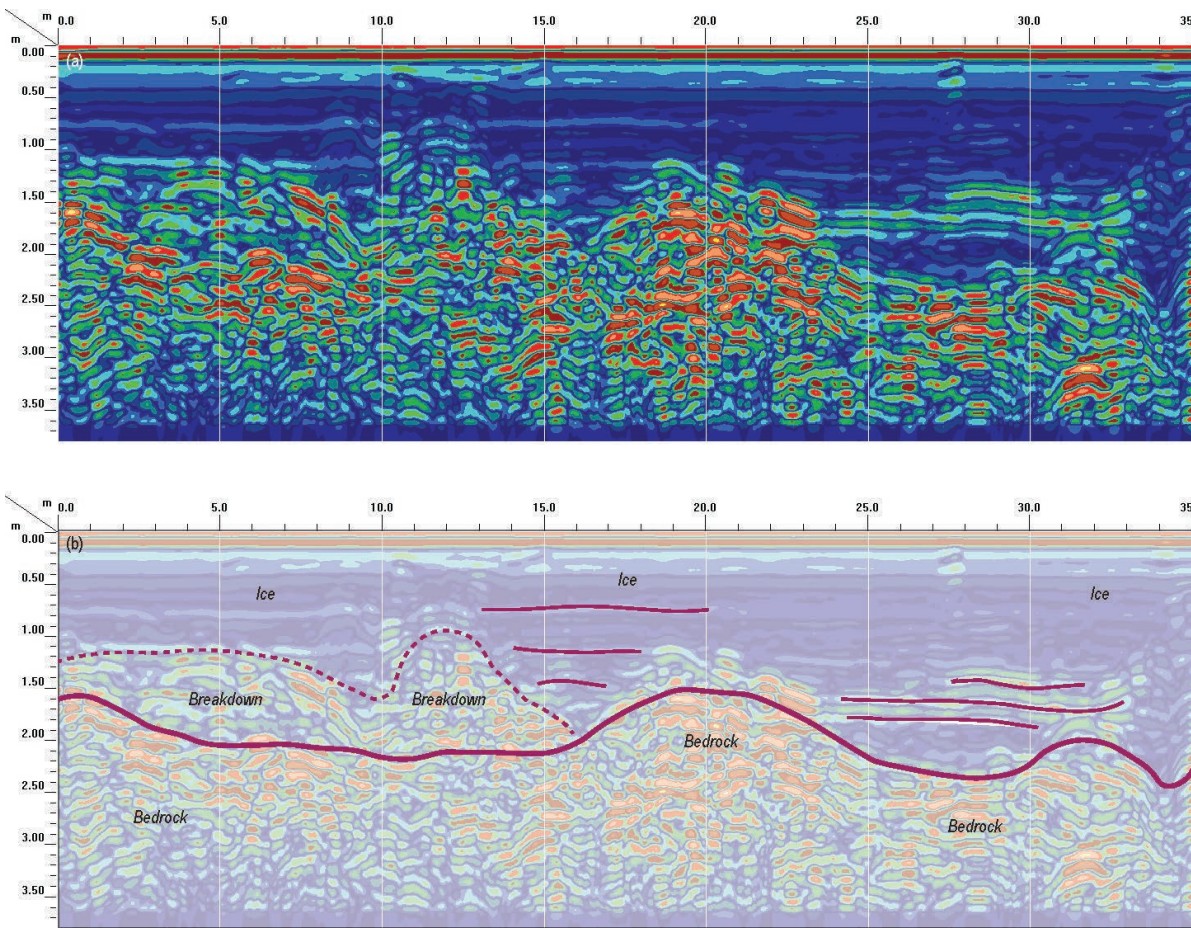

**Figure 4. Results of a Ground Penetrating Radar (GPR) survey along the Frozen Freeway. (a) Uninterpreted GPR data for half of the full survey length, with the back of the cave on the right. Vertical white lines mark measured points every 5 m used for scaling the GPR survey. (b) Interpretation of the GPR survey shown in (a). The bedrock surface is represented by an irregular, but laterally continuous contact. More isolated and discontinuous zones of high amplitude reflectors above this contactor interpreted as piles of breakdown. Ice beneath the Skating Rink, between 20 and 35 m along the transect, reaches a maximum thickness of nearly 2.5 m. Ice along the Frozen Freeway, between 0 and 20 m, has a maximum thickness of ~2 m. Horizontal lines within the ice are likely stratigraphic layers.**





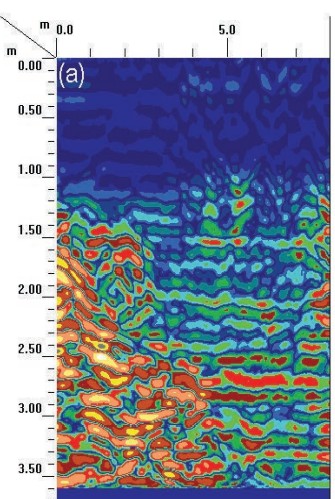 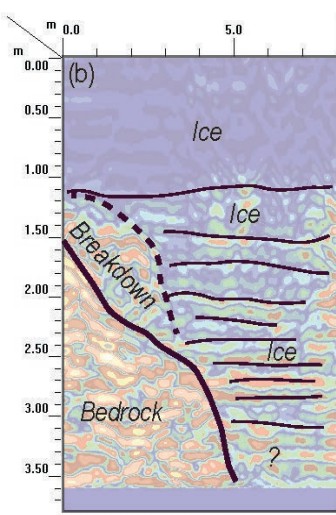

**Figure 5. Results of an 8-m long Ground Penetrating Radar (GPR) across the Icicle Room, presenting uninterpreted (a) and interpreted (b) data. From the surface to a depth of ~1 m the ice is only faintly stratified, whereas below that depth the ice contains increasingly prominent radar reflectors. The steeply sloping bedrock surface is defined on the left side of the transect, mantled in places by breakdown. The maximum thickness of ice here is unclear, but it may be in excess of the 3.5 m penetrated by the 400 MHz GPR system.**




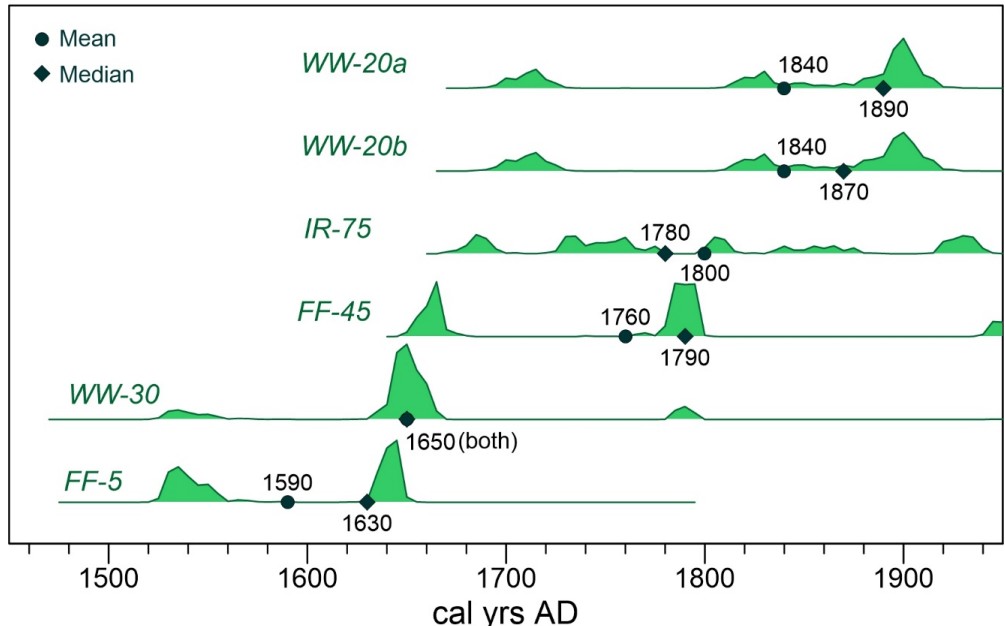


**Figure 6. Calibration ranges from the Intcal 13 calibration curve for radiocarbon dates obtained from Winter Wonderland Cave. Sample labels correspond with Table 1. Circles mark the mean calibrated age for each sample, diamonds denote the median age. Both are labelled in years A.D.**




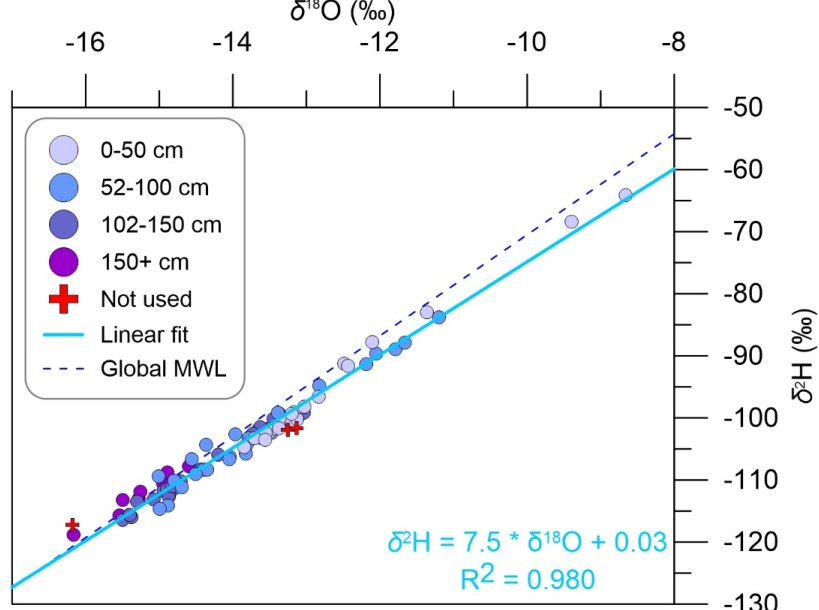


**Figure 7. Plot of δ¹⁸O vs δD for ice samples from the Skating Rink exposure (Fig. 1c). Samples are color-coded to highlight their depth below the top of the exposure. Three samples identified as outliers and discarded from the dataset are marked with red crosses. The solid light blue line represents a linear regression through the remaining 81 samples, and the equation for this line is presented at lower right. The dashed blue line is the global meteoric waterline (Craig, 1961).**


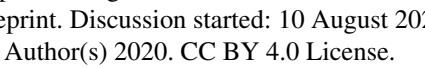



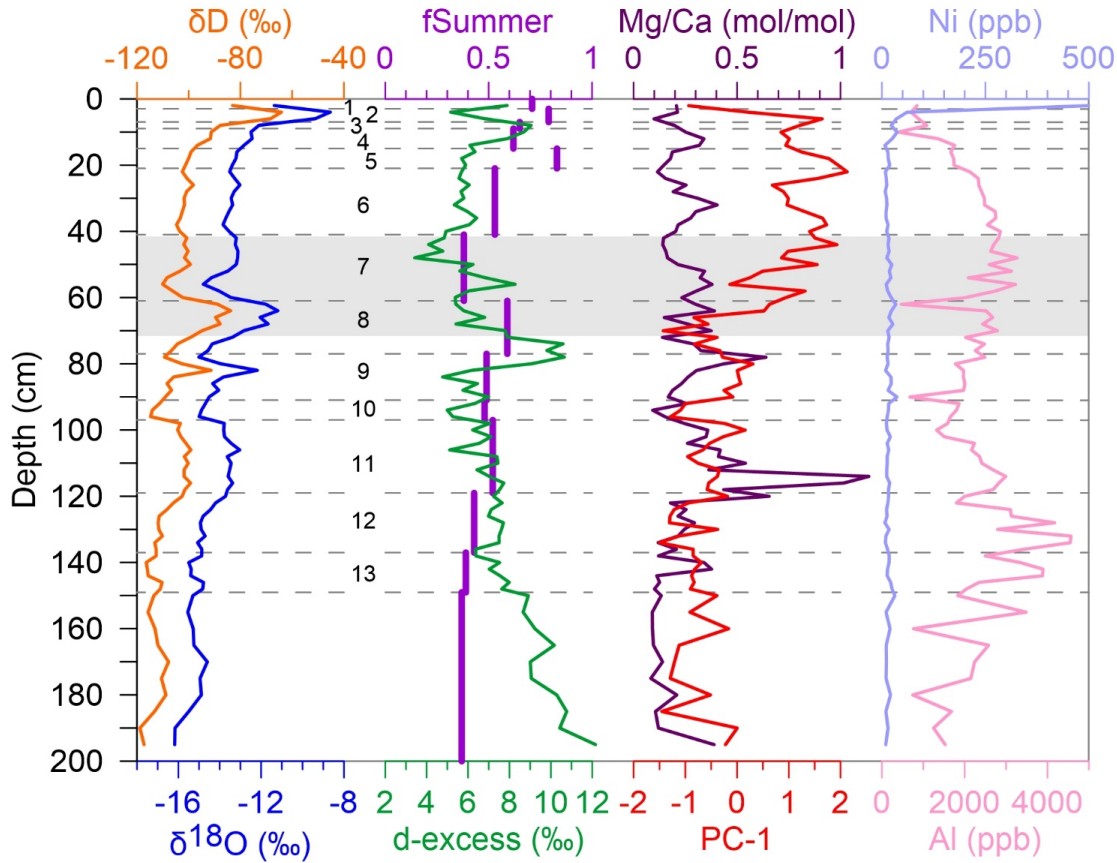

**Figure 8. Composite plot of values measured in the Skating Rink plot against depth below the top of the exposure. Values of δ18O vs δD are shown in the first column. d-excess and the reconstructed fraction of summer monsoonal water in the ice are shown in the second column. The first principle component determined for the glaciochemistry results, along with the molar ratio Mg/Ca are shown in the third column. The two elements which do not fit into PC-1, Al and Ni, are shown in the fourth column. Dashed horizontal lines mark visible layer boundaries observed in the exposure. From top to bottom, these layers are numbered 1 through 13. The shaded gray region marks the approximate depth from which the radiocarbon ages along the Frozen Freeway were obtained, thus the ice in this depth range likely dates to between A.D. 1600 and 1850.**



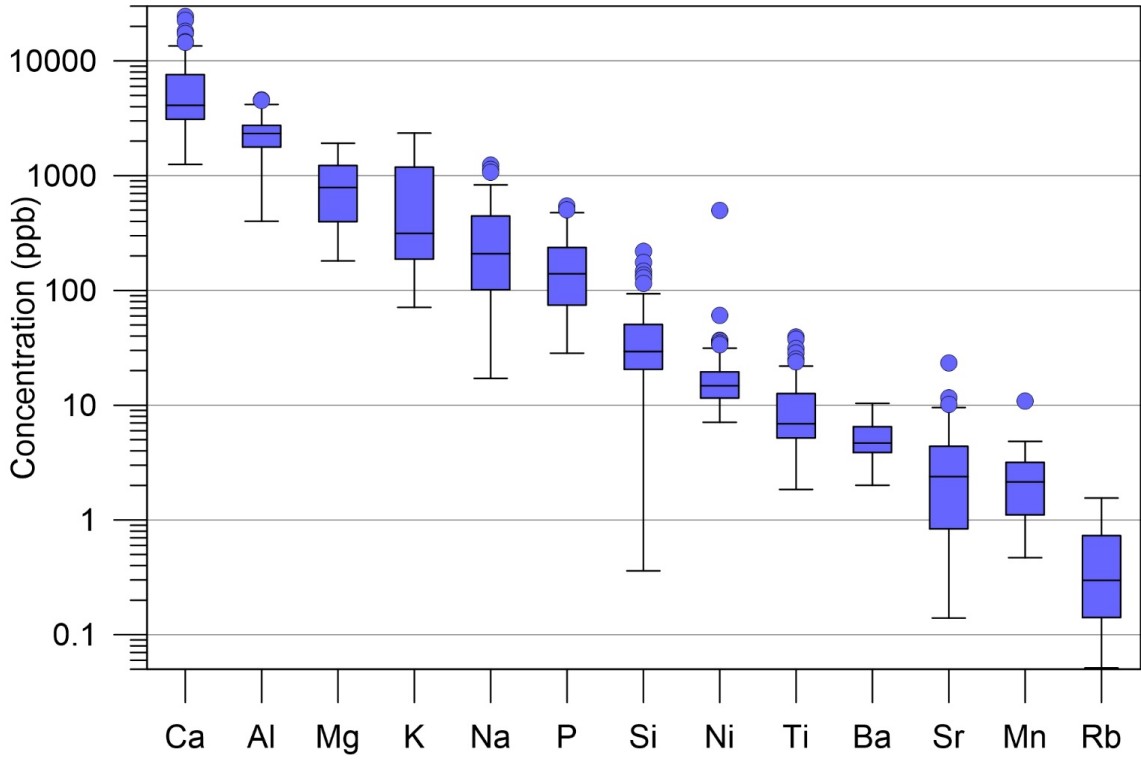


**Figure 9. Box and whisker plot presenting the abundance of detectable elements in samples from the Skating Rink exposure, arranged from most to least abundant. Central lines represent median values, box represents interquartile range, whiskers represent interquartile range ×1.5, and circles represent outliers.**





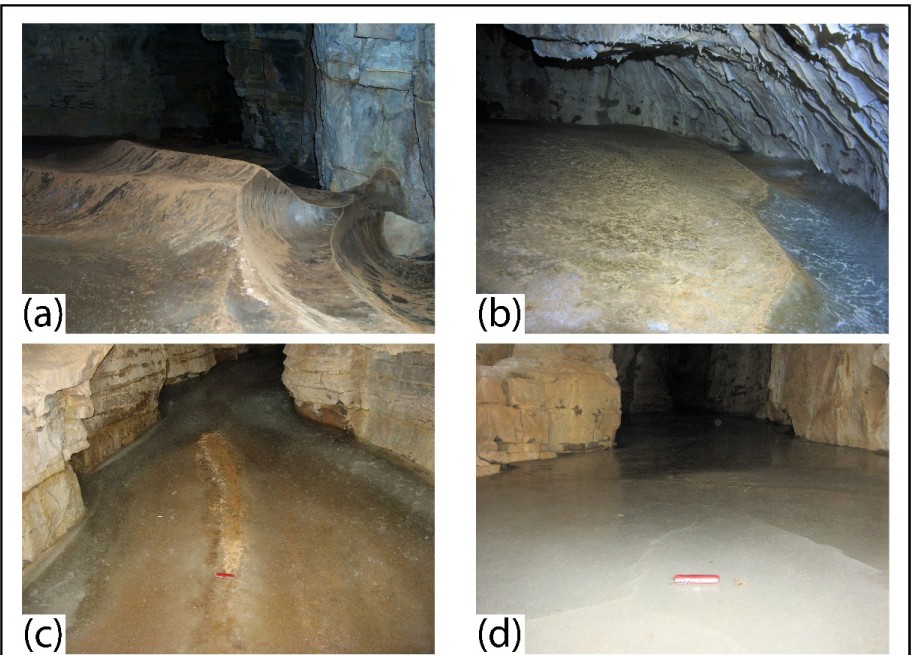


**Figure 10. A selection of photographs highlighting changes observed in Winter Wonderland Cave. (a) The sculpted surface of the ice along the Frozen Freeway in 2016 illustrating numerous intersecting cuspate forms produced by sublimating air currents. The total relief is ~30 cm. The orange material on the ice is cryogenic mineral precipitates. (b) A view showing the edge of the ice in the Skating Rink in 2016 where a ~20-cm deep moat had formed at the contact with the rock wall of the cave. (c) The surface of the Frozen** 795 **Freeway in 2018 illustrating how inflowing water had submerged nearly all of the former sculpted surface under new ice. The red Swiss Army knife is located at the end of a ~1-m long section of the former crest in the ice that still reaches above the new ice surface. (d) Another view from a section of the Frozen Freeway in 2018 where new ice formed from inflowing water has completely inundated the former sculpted ice surface. Red Swiss Army knife is 10 cm long.**



**Table 1. Radiocarbon Dating Results from Winter Wonderland Cave**

| Lab Number -- | Lab -- | Sample -- | Location * | Depth cm | Mass mg | ¹⁴C Age yrs | +/- yrs | δ13C ‰ | Cal Range yrs AD | Mean yrs AD | Median yrs AD | Note -- |
|---|---|---|---|---|---|---|---|---|---|---|---|---|
| 17O/0351 | ICA | WW-20a | FF | 20 | 60 | 40 | 30 | -- | 1690-1730 (20.5), **1810-1920 (74.8)** | 1840 | 1890 | -- |
| 17O/0352 | ICA | WW-20b | FF | 20 | 22 | 50 | 30 | -- | 1690-1730 (21.8), **1810-1920 (73.6)** | 1840 | 1870 | -- |
| OS-128827 | NOSAMS | WWC-IR-75 | IR | 75 | 200 | 145 | 15 | -27.91 | 1670-1700 (15.3), **1720-1780 (33.9)**, 1790-1820 (11.6), 1830-1880 (14.3), 1910-1950 (20.3) | 1800 | 1780 | Composite |
| OS-128826 | NOSAMS | WWC-FF-45 | FF | 45 | 25 | 215 | 15 | -26.71 | 1640-1680 (34.5), **1770-1800 (50.2)**, 1940-x (10.7) | 1760 | 1790 | -- |
| 17O/0353 | ICA | WW-30 | FF | 30 | 18 | 260 | 20 | -- | 1520-1560 (9.4), **1630-1670 (78.2)**, 1780-1800 (7.8) | 1650 | 1650 | Single, intact |
| OS-128804 | NOSAMS | WWC-FF-5 | FF | 5 | 12 | 285 | 12 | -27.26 | 1520-1560 (45.6), **1630-1660 (49.8)** | 1590 | 1630 | -- |
| 17O/0350 | ICA | WW-15 | FF | 15 | 15 | -- | -- | -- | -- | -- | -- | -- |

*FF = Frozen Freeway, IR = Icicle Room





**Table 2. Stable Isotope Results for Winter Wonderland Cave**

|  | ave δ¹⁸O | SD δ¹⁸O | ave dD | SD dD | d-excess | SD d-excess | Slope | Intercept | n | r² | Altered? | Est. In. δ¹⁸O | Est. In. dD | fSummer |
|---|---|---|---|---|---|---|---|---|---|---|---|---|---|---|
|  | ‰ | ‰ | ‰ | ‰ | ‰ | ‰ | -- | ‰ | -- | -- | -- | ‰ | ‰ | -- |
| Layer 1 | -11.36 | -- | -83.02 | -- | 7.88 | -- | -- | -- | 1 | -- | -- | -11.36 | -83.02 | 0.71 |
| Layer 2 | -9.03 | 0.52 | -66.24 | 3.02 | 5.99 | 1.17 | 5.76 | -14.19 | 2 | 1.000 | Y | -10.50 | -75.00 | 0.79 |
| Layer 3 | -12.10 | -- | -87.77 | -- | 9.05 | -- | -- | -- | 1 | -- | -- | -12.10 | -87.77 | 0.65 |
| Layer 4 | -12.59 | 0.21 | -93.12 | 2.98 | 7.57 | 1.34 | 13.70 | 79.24 | 3 | 0.965 | Y | -12.40 | -90.00 | 0.62 |
| Layer 5 | -13.26 | 0.12 | -100.11 | 1.10 | 5.97 | 0.33 | 9.17 | 21.48 | 3 | 0.924 | Y | -10.00 | -70.00 | 0.83 |
| Layer 6 | -13.46 | 0.23 | -101.94 | 1.81 | 5.75 | 0.42 | 7.64 | 0.90 | 10 | 0.949 | N | -13.46 | -101.94 | 0.53 |
| Layer 7 | -13.62 | 0.59 | -103.42 | 3.62 | 5.55 | 1.39 | 5.98 | -21.94 | 10 | 0.961 | Y | -15.20 | -112.00 | 0.38 |
| Layer 8 | -12.73 | 1.27 | -94.40 | 8.28 | 7.44 | 1.97 | 6.51 | -11.56 | 8 | 0.996 | Y | -13.75 | -100.00 | 0.51 |
| Layer 9 | -13.98 | 0.89 | -104.74 | 6.35 | 7.13 | 2.05 | 6.88 | -8.52 | 7 | 0.921 | N | -13.98 | -104.74 | 0.49 |
| Layer 10 | -14.86 | 0.02 | -113.31 | 1.85 | 5.53 | 0.71 | 12.06 | 65.05 | 3 | 0.959 | Y | -14.00 | -105.00 | 0.48 |
| Layer 11 | -13.55 | 0.23 | -101.54 | 1.60 | 6.87 | 0.75 | 6.47 | -13.92 | 11 | 0.826 | N | -13.55 | -101.54 | 0.52 |
| Layer 12 | -14.63 | 0.43 | -109.73 | 3.50 | 7.30 | 0.43 | 8.01 | 7.45 | 9 | 0.985 | N | -14.63 | -109.73 | 0.43 |
| Layer 13 | -15.13 | 0.33 | -113.66 | 2.71 | 7.35 | 0.57 | 8.03 | 7.73 | 6 | 0.956 | N | -15.13 | -113.66 | 0.39 |
| Bottom | -15.36 | 0.54 | -113.05 | 3.74 | 9.87 | 1.11 | 6.69 | -10.32 | 10 | 0.940 | N | -15.36 | -113.05 | 0.37 |
| Mean All | -13.95 | 1.38 | -103.94 | 9.94 | 7.68 | 2.46 | 7.03 | -5.85 | 94 | 0.957 | -- | -- | -- | -- |
| Mean SR* | -13.80 | 1.30 | -103.35 | 9.75 | 7.05 | 1.69 | 7.40 | -1.26 | 84 | 0.976 | -- | -- | -- | 0.55 |
| Mean IR* | -15.23 | 1.45 | -108.84 | 10.69 | 12.98 | 1.25 | 7.36 | 3.20 | 10 | 0.994 | -- | -- | -- | -- |

*SR = Skating Rink, IR = Icicle Room