# Peer review of "First Investigation of Perennial Ice in Winter Wonderland Cave, Uinta Mountains, Utah, USA"

_The Cryosphere, 2020_

## Referee Comment (RC1) · Aurel Perşoiu (Referee) · 26 Sep 2020

Over the past decade, investigations of perennial cave ice deposits took a more central stage, as the continuous rise in temperatures (and associated climatic changes) threaten these rather understudied components of the cryosphere. In this context, understanding the genesis and behavior of cave ice deposits could lead to the development of novel proxies of past climate variability that could add unique insights in past climate variability. Munroe attempts to do so by applying a range of investigation tools to several small perennial ice deposits in a cave in the Rocky Mountains, Utah and presenting and discussing the preliminary results. While the paleoclimatic potential of the investigated cave ice deposits is rather small, the results could advance our general knowledge of ice cave processes. While I had numerous comments, I never-

theless think that the manuscript could be published, given that 1) the paleoclimatic reconstruction is diminished in importance (see the comments on chronology) and 2) processes in the cave are emphasized as a tool for subsequent studies.

General comments One of the main problems of the paper is the chronology. It is not clear at all what was actually dated. My understanding is that the top 15 cm of ice were dated. Because of the uneven ablation of the cave surface this approach is extremely problematic when it is intended to be used for climate reconstructions. If I understood correctly, the surface of the ice is extremely uneven and samples were collected from different depths, measured ("estimated visually" – a rather unusual choice) against the nearest ridge. Sublimation and/or melting was definitely acting with different intensities over the surface of the ice, bot today and in the past, and as such, younger packrat droppings could have been incorporated in older ice and or older ones reworked and deposited in new ice. Based on these considerations, I think the only message that can be obtained from the 14C dating is that ice was present in the cave between the youngest and oldest dates. Anyway, sampling locations need to be marked on the cave map. A detailed sketch of the ice surface with the position of the dated samples against morphology (and stratigraphy, if available) will help us understand what was being done.

I am surprised that no attempt was made to collect and use bulk electromagnetic wave propagation velocities – these could have been used to peek into the composition of the ice (a method used by Hausmann & Behm, 2011, which is cited by the author). Is the data available and usable? Even if not "fantastic", it could further help subsequent studies (elsewhere). The point I want to make (here and through the review) is to have as much as possible methods descriptions and data available, rather than only "publishable" ones.

Specific comments The final paragraph of the introduction reads like being taken straight from a research proposal. Perhaps it should be rephrased to sound more article-like. Field site: it would benefit the readers to add one line about the characteristic of the limestone (rather than the name of the formation which would be in negligible interest). E.g., primary/secondary porosity and thickness of caprock are important to understand how water reaches the cave and/or how heat is being transferred to the cave. Also, for the understanding of ice chemistry, the general lithology of the rock should be presented.

Line 89: at what height above the ice were the loggers suspended and at what distance from the rock walls? These are important considerations for the understanding of cave meteorology and factors leading to ablation/formation of ice, as the both presence of ice and of air currents induce strong vertical thermal gradients.

Line 100: please detail the "variety of gain settings" that were used during the GPR data acquiring (later in the manuscript, values are mentioned) and discuss the choice of one over the other. This is important if this study is to be useful for other researchers.

Line 115-126: see my general comment. What calibration curve was used? The most recent is Raimer et al., 2020.

Line 129-132: please describe the stratigraphy of the exposure and the number of samples collected per layer as the results are later presented using the layering. Did you consider layering during sampling or cut across strata (as suggested by the 2 cm spacing)? As water formed by the freezing of water, fractionation would have resulted in different stable isotope values within the same layer of ice.

Line 134-136: please detail the location of the additional samples for stable isotope analyses.

Line 159: see my comment on the height of data loggers above cave ice surface.

Line 167: using freezing degree days is rather uncommon, some readers might think that the number of days was calculated, rather than the sum of degrees below 0 °C. Please define it in the text. Perhaps the fdd should be calculated for the cave data, as well.

Line 212-217: these should be moved under "methods"

Line 219-220: move to methods.

Line 220: "depleted" against…? Generally speaking, values cannot be depleted. A sample can be depleted in the heavy (light) isotope, resulting in a low (high) delta value.

Line 224-229: I do not understand the reason for removing "outliers", the wide range of values is not an issue. This is a very unusual approach; I would consider all results, especially as the omitted results are almost similar to the ones used (e.g., fig. 7)

Line 245: relative to…?

Line 254-255: move to methods, and detail

Line 259: Ni – possible contamination of the upper layers?

Line 274-276: I do not understand the rationale behind the explanation for the chimney. If outside air gets below the internal one, cold air will flow inside the cave, regardless of the presence or absence of a chimney (perhaps references in lines 2776-277 should be updated). Form the data in gig. 2, I do not see the need for a secondary entrance. The morphology of the cave and the data clearly indicates a "cold air trap", with dynamic cooling as cold air flows inside the cave and slow warming, perhaps triggered by geothermal heat (and additional heat brought to the cave by dripwater). This does not exclude the presence of a chimney, but if it was not observed during the visits, perhaps the simplest explanation would be sufficient (Occam's razor).

Line 284-294: elegant

Line 292: did you notice airflow? Perhaps it is just warming propagating outwards though conduction from the warmer inner parts of the cave.

Line 295: what you mean by "back"? The longitudinal profile indicates flow from the entrance towards the "back" of the cave.

Line 299: again, was there a chimney/conduit observed? If not, water will just drip through the limestone's fissures.

Line 302: no need to flood the epikarst (it would be problematic at that altitude); in limestone, water will always find a way towards lower altitudes/caves.

Line 303: it would be useful to discuss the "ridges and troughs" on the surface of ice in detail. Inflow of warm water would definitely result in melting of ice, leading to the formation of surface micro topography. Further, the same water could bring in young packrat droppings and redeposit them in older ice, exposed during melting. This is evident in figure 10, where ridges are drowned in newly formed ice.

Line 304: not seeing a secondary entrance, only hypothesize one, I would consider the cave a "static cave with congelation ice" (Luetscher and Jeannin, 2004a)

Line 318: Is there possibly ice present under the breakdowns (in the "ice free" sections)? Perhaps a continuous layer of ice extends from the entrance through the cave and the breakdowns cover it in places. It would be otherwise difficult to understand the lack of ice in parts of the cave.

Line 324-326: the low reflectivity could indicate a thick and homogenous layer of clear ice, formed by the slow freezing of water in a through between ridges (so-called "lake ice"). And indeed, this ice would be free of cryogenic calcite and other sediments, that would settle at the bottom of the lake water during freezing. Here the usage of use bulk electromagnetic wave propagation velocities could help. Line 334: a photo and stratigraphic sketch would help understand the structure of the ice deposit and the stable isotope values

Line 342: depleted in heavy isotopes

Line 353 and subsequent paragraph: "freezing slopes" have been described as being generally below 7.2 (Jouzel and Souchez, 1982, Souchez and Jouzel, 1984, Souzhez et al., 2000, Persoiu et al., 2011). Layers 4, 5, 6, 10, 12 and 13 (but not 9 and 10, which

have slopes below 7) likely formed as thin layers of water froze on top of the existing ice block (slope>7). This could be checked by plotting d-exces vs. d2H (Souzchez et al., 2000) for every layer. Lack of correlation between the two parameters would indicate kinetic conditions and thus open-system freezing (thin layers of water freezing on top of existing ice). Alternatively, all layers could have formed as "lake ice" and subsequently part of them melted away thus resulting in the loss of the alignment along a line with a slope below 7.

Line 363: technically, it was the isotopologues that were fractionated...

Line 364-366: because during freezing of a pool of water the samples align one a straight line in a dH-$\delta$18O diagram (with r2>0.9), potential loss of top and/or bottom samples would not affect to much the slope of the line.

Line 371-372: regardless of type of freezing, fractionation occurs. It is the type of freezing and fractionation that matters. In layers of ice formed by the freezing of thin films of water, all water freezes "at once" and as such the stable isotope composition of the resulting ice is similar to that of the water. In the case of freezing of a pool of water, fractionation and continuous incorporation of heavy isotopes in ice would result in stable isotope trend from top to the bottom. If the entire layer is sampled, the stable isotope composition of ice is similar to that of the parent water; but if samples are collected at various depths, their stable isotope composition would differ (higher values at top, lower at the bottom). Further, partial melting would result in loss of usually top layers, enriched in heavy isotopes, so that the stable isotope values of the ice would have little resemblance of that of original water. Elegantly, and nicely employed here, intersecting the LMWL with the slope of samples from every such layer would result in the original water before freezing (regardless of ice loss, freezing intensity, fractionation factors etc). This intersection should be applied to layers with low values of the slope (2, 7, 8, 9, 11 and bottom) and one single data point per layer should be than plotted in fig. 8 (assuming thus that each of these layers formed from one single pool of water – as for example, the one in Fig. 10d). However, this would alter the discussion in lines

374-384.

Line 380: I believe it is assumed here that the delta values reflect long-term annual means, in which case the difference is indeed large, pointing towards extreme climate shifts. Alternatively, it could be that they reflect different recharge patterns (e.g., winter vs. summer recharge).

Lines 379-384: in the absence of age control, I would refrain form discussing LGM-old ice. Further, it would be difficult to explain the survival of ice in a small cave, with such a dynamic ice accumulation/ablation processes.

Lines 396-404: this could be somewhat shortened to a line, not to break the stable isotope discussion.

Lines 404-412: perhaps a lengthier discussion of stable isotope variability in winter vs. summer could be included, with data from other stations nearby. Two data points are not enough to sustain the subsequent modeling. Further, if OIPC is used for summer, it should also be used for winter (especially as the OIPC-derived data for winter is depleted in 18O by about 3 ‰ compared to measured values). So, either used OIPC only, or data from stations, but not a combination of the two. I understand that this would affect the modeling in lines 412-420, but it is more correct.

Lines 422-423: I still cannot see the reconstructed values.

Lines 422-434: see my comment above and perhaps redo the calculations (using one value for the layers formed as "lake ice")

Lines 435-442: I find the discussion in this paragraph difficult to sustain by the data. Given the complex morphology of the ice surface and the lack of stratigraphic sketches, it is difficult to follow.

Lines 444-456: in the absence of chronology, this section is very speculative and could be safely left out.

[Figure]

"Glaciochemistry interpretation" should be moved before paleoclimate and "indication of recent change" after the climate data. However, in the absence of chronology, the entire discussion of the potential sources of variability in the chemistry of ice is speculative. As above, it could be reduced to a few lines, detailing the layer-by-layer variability, rather than potential temporal variability. For instance, are there differences between the two types of ice (lake vs. floor) in terms of chemistry? This could be more helpful for subsequent studies of cave ice deposits with better age control.

Line 469: see my comments on chronology. Ages should not be employed when discussing the layers, as 1) it was only the surface that was dated and 2) complex surface morphology, with deep ridges.

"Limitations and direction for future research" and "conclusions" These two chapters should me merged and emphasize should be put on the potential usage of data obtained from this case in the general understanding of cave ice processes.

The first panel in figure 2 is slightly misleading du to the inverted scale, please use a normal one (and perhaps write the "warmer outside, warmer in the cave" text with the same color as that of the line.

Souchez, R., Jouzel, J., Lorrain, R., Sleewaegen, S., Stiévenard, M., and Verbeke, V.: A kinetic isotope effect during ice formation by water freezing, Geophysical Research Letters, 27, 1923-1926, 10.1029/2000GL006103, 2000.

---

## Referee Comment (RC2) · Marc Luetscher (Referee) · 14 Oct 2020

The author presents a detailed description of a recently discovered mid-latitude/high altitude ice cave located in the Uinta Mts, Utah (USA). The preliminary results suggest the cave ice is rather young and likely associated with enhanced winter snow precipitation during the Little Ice Age. Recent ingress of liquid water hints towards partial thawing of this sporadic mountain permafrost which is consistent with the general warming trend observed regionally.

The paper is nicely written and presents original data supporting the site-specific dynamics of the system. Such documentation is critical to better understand the response of the subsurface cryosphere with respect to climate change and provides fun-

damental data on a largely under-explored environment. Yet, the paper contributes only marginally to a better understanding of the general processes controlling cave ice mass balances nor does it provide a significant paleoenvironmental record. Therefore, I strongly suggest to emphasize the significance of this site for the larger community by stressing out its specificity with respect e.g. to other ice caves and/or its significance for regional mountain permafrost. Similarly, the discussion identifies a range of tasks (§4.5) that could be undertaken during future field work. Rather, I'd recommend focusing/rephrasing this chapter on what should be done at this site to answer key questions associated with ice caves, mountain permafrost and/or paleoenvironmental reconstructions. Providing these few adjustments and some minor comments below, I fully support publication of this paper.

Detailed comments:

L. 16: edit "Most values of $\delta$18O and $\delta$D range plot subparallel" (delete "range")

L. 17: edit "with a slope of 7.5 with and an intercept at 0.03‰' (delete "with")

L. 61-66: redundancy with l. 83-85.

L. 88: specify the location of the loggers

L. 90: what about the loggers' calibration?

L. 115: specify the location of your samples, in particular with respect to a cave ice stratigraphy and location of the isotope samples

L. 160: did you mean r2? Check significant digits

L. 194: can you comment about the wave propagation speed?

L. 219: I am missing some info about the modern context. What is the temperature mean and range, resp. the isotope signature, in the outside environment?

L. 220: "both isotopes are notably depleted" I guess you mean that both isotope ratios

are low, i.e. depleted in the heavy isotopes (?)

L. 221-222: check significant digits on the isotope values

L. 272 "drops notably below ∼0°C" or below the cave air temperature?

Fig. 8 check/remove unit of Mg/Ca

Table 1 I guess it is the sample mass and not the C-mass? Specify which calibration curve was used.

Table 2 check the significant digits; the precision is given at ±0.2‰ and ±1.0‰ for d18O and d2H, respectively (cf. l. 143)

---

## Author Comment (AC1) · 2 Nov 2020

Over the past decade, investigations of perennial cave ice deposits took a more central stage, as the continuous rise in temperatures (and associated climatic changes) threaten these rather understudied components of the cryosphere. In this context, understanding the genesis and behavior of cave ice deposits could lead to the development of novel proxies of past climate variability that could add unique insights in past climate variability. Munroe attempts to do so by applying a range of investigation tools to several small perennial ice deposits in a cave in the Rocky Mountains, Utah and presenting and discussing the preliminary results. While the paleoclimatic potential of the investigated cave ice deposits is rather small, the results could advance our general knowledge of ice cave processes. While I had numerous comments, I nevertheless think that the manuscript could be published, given that 1) the paleoclimatic reconstruction is diminished in importance (see the comments on chronology) and 2) processes in the cave are emphasized as a tool for subsequent studies.

Thank you Aurel for the obvious time and consideration you put into reviewing my manuscript *"First Investigation of Perennial Ice in Winter Wonderland Cave, Uinta Mountains, Utah, USA"*. I always learn a lot about ice caves and stable isotopes when I talk with you, and it is extremely helpful to have your questions, suggestions, and supportive criticisms in hand as I contemplate how to improve the manuscript. I am encouraged to see that you agree that Winter Wonderland Cave (WWC) offers a unique opportunity to study the processes responsible for accumulation of ice in caves, and the ways in which this ice can retain information about paleoclimate. Here I summarize the changes I plan to make in response to your review.

General comments: One of the main problems of the paper is the chronology. It is not clear at all what was actually dated. My understanding is that the top 15 cm of ice were dated. Because of the uneven ablation of the cave surface this approach is extremely problematic when it is intended to be used for climate reconstructions. If I understood correctly, the surface of the ice is extremely uneven and samples were collected from different depths, measured ("estimated visually" – a rather unusual choice) against the nearest ridge. Sublimation and/or melting was definitely acting with different intensities over the surface of the ice, bot today and in the past, and as such, younger packrat droppings could have been incorporated in older ice and or older ones reworked and deposited in new ice. Based on these considerations, I think the only message that can be obtained from the 14C dating is that ice was present in the cave between the youngest and oldest dates. Anyway, sampling locations need to be marked on the cave map. A detailed sketch of the ice surface with the position of the dated samples against morphology (and stratigraphy, if available) will help us understand what was being done.

I agree that it would be great to have better control on the age of this ice, but that wish is not unique to WWC: as I note in the manuscript, dating subterranean ice is rarely

straightforward or easy.  That's why it is so significant that organic matter was recovered from the ice in WWC and radiocarbon dated.  These packrat droppings were from within the ice, not the ice surface, and are not concentrated in a horizontal layer in association with cryogenic minerals.  Thus, it seems unlikely that they represent older material which accumulated as a lag during sublimation.  At the same time, there is no realistic mechanism for incorporating younger packrat droppings into older ice.  Thus, the most straightforward interpretation of these ages is that they faithfully represent the age of the ice around them.  Yes, the surface of the ice in the main part of the cave has been affected by sublimation and sculpted into a series of ridges and valleys.  What I should have made clearer in the manuscript is the relationship between that ice surface, the strata from which the samples were collected, and the overall stratigraphy of the ice body.  I did mention in the text that the ice locally reaches to the cave ceiling, whereas most of the floor of the Frozen Freeway (where the organic samples were collected) is actually a lower stratigraphic level.  I also highlighted this with the shading in Figure 8, and explained this in the caption (Lines 782-783).  However, I can understand that this is perhaps difficult to visualize.  Therefore, I plan to add a new figure that will display the stratigraphy of the ice, the relationship between the Frozen Freeway surface and the ultimate uppermost layer of the sampled ice exposure, and the stratigraphic level where the samples were collected for radiocarbon dating.  I'm confident that this will give the reader a better understanding of the real strength of the age control available for this ice.  Looked at another way, nearly 2 m of ice is present below the radiocarbon ages; this ice must be older than ~AD 1600.  Similarly, about half a meter of ice is present (extending to the cave ceiling) above the youngest radiocarbon ages.  I would argue that this ice accumulated in the last century.

> I am surprised that no attempt was made to collect and use bulk electromagnetic wave propagation velocities – these could have been used to peek into the composition of the ice (a method used by Hausmann & Behm, 2011, which is cited by the author). Is the data available and usable? Even if not "fantastic", it could further help subsequent studies (elsewhere).  The point I want to make (here and through the review) is to  have as much as possible methods descriptions and data available, rather than only "publishable" ones.

The suggestion that bulk EM wave propagation velocities be investigated is a good one.  As I note in the section on directions for future work, the investigations reported in this manuscript do not represent the end of fieldwork in this cave.  Future geophysical investigations would certainly be useful for further clarifying three-dimensional architecture of the ice.

> Specific comments The final paragraph of the introduction reads like being taken straight from a research proposal. Perhaps it should be rephrased to sound more article-like. Field site: it would benefit the readers to add one line about the characteristic of the limestone (rather than the name of the formation which would be in negligible interest). E.g., primary/secondary porosity and thickness of caprock are important to understand how water reaches the cave and/or how heat is being transferred to the cave. Also, for the understanding of ice chemistry, the general lithology of the rock should be presented.

I will definitely add more details about the lithology of the Madison Limestone so that the reader can better understand the properties of the host rock.

> Line 89: at what height above the ice were the loggers suspended and at what distance from

the rock walls? These are important considerations for the understanding of cave meteorology and factors leading to ablation/formation of ice, as the both presence of ice and of air currents induce strong vertical thermal gradients.

Certainly I can provide more information about how the data loggers were suspended from the ceiling – how far from the ice and how far from the walls they were located. I think it is important to emphasize that they were suspended however; they were not deployed directly on the ice surface. This allows them to better capture actual changes in air temperature.

Line 100: please detail the "variety of gain settings" that were used during the GPR data acquiring (later in the manuscript, values are mentioned) and discuss the choice of one over the other. This is important if this study is to be useful for other researchers.

It seemed simpler to mention that a range of settings was used because many different radar transects were collected within the cave. These settings were determined through an auto gain function employed each time the antenna was placed on the ice surface at the start of a new transect. I will clarify this in my revisions.

Line 115-126: see my general comment. What calibration curve was used? The most recent is Raimer et al., 2020.

The IntCal13 curve was used, as was mentioned in Line 124 and was the newest available at the time the manuscript was written. There is very little difference in these young calibration ranges when recalibrating with Intcal 20.

Line 129-132: please describe the stratigraphy of the exposure and the number of samples collected per layer as the results are later presented using the layering. Did you consider layering during sampling or cut across strata (as suggested by the 2 cm spacing)? As water formed by the freezing of water, fractionation would have resulted in different stable isotope values within the same layer of ice.

Details about the stratigraphy of the exposure in the sampling strategy were presented in Lines 240 to 249, the visual divisions between the layers are plotted in Figure 8, and the number of samples per layer is stated in Table 2. Nonetheless, I will include this information in the new figure that I am planning, as explained above in my response to your comment about the overall chronology.

Line 134-136: please detail the location of the additional samples for stable isotope analyses.

These will be added to the main location figure.

Line 159: see my comment on the height of data loggers above cave ice surface.

Line 167: using freezing degree days is rather uncommon, some readers might think that the number of days was calculated, rather than the sum of degrees below 0 °C. Please define it in the text. Perhaps the fdd should be calculated for the cave data, as well.

I'll be sure to define the freezing degree day calculation.

Line 212-217: these should be moved under "methods"

I will make this change.

Line 219-220: move to methods.

I'm not sure I agree – talking about the total number of samples collected seems more like a result to me.

Line 220: "depleted" against. . .? Generally speaking, values cannot be depleted. A sample can be depleted in the heavy (light) isotope, resulting in a low (high) delta value.

Good point, I will clarify that statements like "depleted" or "enriched" are relative to the standard.

Line 224-229: I do not understand the reason for removing "outliers", the wide range of values is not an issue. This is a very unusual approach; I would consider all results, especially as the omitted results are almost similar to the ones used (e.g., fig. 7)

Screening to identify outliers in the isotope data seemed appropriate given the possibility that some of these ages have been greatly affected by fractionation. In applying this approach I was following the work of Benjamin et al. (2005[1]), who screened their isotope data similarly before creating the local meteoric water line that I reference in my analysis. Perhaps this step is overly conservative, but I do think is important to evaluate the data before launching into analysis and interpretation. In any event, only three of the samples were eliminated by this screening, so my analysis essentially moves forward with the data set intact.

Line 245: relative to. . .?

Line 254-255: move to methods, and detail

Line 259: Ni – possible contamination of the upper layers?

I don't think so, because there is no source of Ni in the host rock. And as for contamination in the laboratory, all of the samples are treated identically and all were run sequentially in the same batch on the ICP-MS.

Line 274-276: I do not understand the rationale behind the explanation for the chimney. If outside air gets below the internal one, cold air will flow inside the cave, regardless of the presence or absence of a chimney (perhaps references in lines 2776-277 should be updated). Form the data in gig. 2, I do not see the need for a secondary entrance. The morphology of the cave and the data clearly indicates a "cold air trap", with dy- namic cooling as cold air flows inside the cave and slow warming, perhaps triggered by geothermal heat (and additional heat brought to the cave by dripwater). This does not exclude the presence of a chimney, but if it was not observed during the visits, perhaps the simplest explanation would be sufficient (Occam's razor).

In my revisions I will provide additional explanation for why I think there is an inaccessible chimney connecting the rear of the cave with the plateau surface above. Yes, in the winter cold air will sink into the cave regardless of the number of entrances. But, as I mention in the text, the rear of the cave drafts strongly outward in the summer. And during each visit, a strong jet of air was noted passing outward through the cave entrance. This volume of air, combined with the sculpted surface of the ice in the Frozen Freeway, implies that a large amount of air is moving through the cave system. Given the distribution of temperatures measured at the three loggers, and the changes in these temperatures during the year, it seems likely that there is a conduit through which air can reach the back of the cave.
* * *
[1]Benjamin, L., Knobel, L.L., Hall, L.F., Cecil, Ld., and Green, J.R., 2005, Development of a local meteoric water line for southeastern Idaho, western Wyoming, and southcentral Montana: http://pubs.usgs.gov/sir/2004/5126/.

Line 284-294: elegant

Thank you!

Line 292: did you notice airflow? Perhaps it is just warming propagating outwards though conduction from the warmer inner parts of the cave.

Yes, this will be emphasized in the revision.

Line 295: what you mean by "back"? The longitudinal profile indicates flow from the entrance towards the "back" of the cave.

The observations in the summers of 2018 & 2019 clearly indicate that water entered from the rear of the cave, flooding out over the ice surface and reaching about halfway along the Frozen Freeway.

Line 299: again, was there a chimney/conduit observed? If not, water will just drip through the limestone's fissures.

I will clarify this in the revisions, but the only liquid water that we encountered was in the rear of the cave, and the closer we got to the ultimate back of the cave, the wetter the ice surface became. It is very clear that the water is entering from the rear of the cave, rather than dripping from multiple points in the ceiling.

Line 302: no need to flood the epikarst (it would be problematic at that altitude); in limestone, water will always find a way towards lower altitudes/caves.

True, I will reword this in the revision.

Line 303: it would be useful to discuss the "ridges and troughs" on the surface of ice in detail. Inflow of warm water would definitely result in melting of ice, leading to the formation of surface micro topography. Further, the same water could bring in young packrat droppings and redeposit them in older ice, exposed during melting. This is evident in figure 10, where ridges are drowned in newly formed ice.

There is no sign that the ridges and troughs are produced by warm water, rather they appear to have been sculpted by airflow. Certainly water flowing over the surface of the ice could we distribute cryogenic minerals and organic matter, but it would not redeposit this material within "older ice"; rather this process would create a situation where older material is encased in younger ice. That material would be concentrated in a single stratigraphic level, clearly indicating its origins through melt out related concentration. That is not the case with the organic matter dated in this study, as I noted above.

Line 304: not seeing a secondary entrance, only hypothesize one, I would consider the cave a "static cave with congelation ice" (Luetscher and Jeannin, 2004a)

Again, given the clear evidence for strong airflow through the cave in the form of the sculpted ice surface, and the fact that strong drafting was noted during each field visit, I am inclined to lean toward the interpretation of hypothesized but currently inaccessible second entrance. Nonetheless, I will add wording to the revision to indicate the hypothesize nature of this conduit.

Line 318: Is there possibly ice present under the breakdowns (in the "ice free" sec- tions)? Perhaps a continuous layer of ice extends from the entrance through the cave and the breakdowns cover it in places. It would be otherwise difficult to understand the lack of ice in parts of the cave.

Possibly, but this cannot be proven or disproven from the available evidence. In reality I think that the central and forward part of the cave where the breakdown is concentrated probably represent a mixing zone between ice derived from water

entering the rear of the cave and flowing forward, and ice derived from water that runs into the Icicle Room through the entrance.

> Line 324-326: the low reflectivity could indicate a thick and homogenous layer of clear ice, formed by the slow freezing of water in a through between ridges (so-called "lake ice"). And indeed, this ice would be free of cryogenic calcite and other sediments, that would settle at the bottom of the lake water during freezing. Here the usage of use bulk electromagnetic wave propagation velocities could help. Line 334: a photo and stratigraphic sketch would help understand the structure of the ice deposit and the stable isotope values

This is a good point, however looking at the current surface of the ice where it has been sculpted by the airflow reveals a nearly continuous layer of concentrated cryogenic minerals. These would form a strong reflector in GPR data. Furthermore, the scale of these ridges and troughs is on the order of 10 to 30 cm, whereas the scale of the apparently reflector-free ice seen in the GPR data is on the order of the meter. If continuous, meter deep pools of water were present on the surface of the ice at some point in the past, then yes they could have frozen as a lake ice creating a GPR record like the one that I imaged. However, this would require ridges and troughs on a scale quite different than that observed in the 2018 ice surface.

> Line 342: depleted in heavy isotopes

> Line 353 and subsequent paragraph: "freezing slopes" have been described as being generally below 7.2 (Jouzel and Souchez, 1982, Souchez and Jouzel, 1984, Souzhez et al., 2000, Persoiu et al., 2011). Layers 4, 5, 6, 10, 12 and 13 (but not 9 and 10, which have slopes below 7) likely formed as thin layers of water froze on top of the existing ice block (slope>7). This could be checked by plotting d-exces vs. d2H (Souzchez et al., 2000) for every layer. Lack of correlation between the two parameters would indicate kinetic conditions and thus open-system freezing (thin layers of water freezing on top of existing ice). Alternatively, all layers could have formed as "lake ice" and subsequently part of them melted away thus resulting in the loss of the alignment along a line with a slope below 7.

As shown in Table 2, I calculated the slopes of regressions through the stable isotope data for each layer as a way to identify layers that were likely impacted by extensive fractionation during closed-system (lake ice) freezing. These values are presented in the "slope" column of Table 2. Layers noted as "Y" in the "Altered" column of Table 2 are those with slopes suggesting extensive fractionation of some other process, and were corrected as noted below.

> Line 363: technically, it was the isotopologues that were fractionated. . .

> Line 364-366: because during freezing of a pool of water the samples align one a straight line in a dH-$\delta$18O diagram (with r2>0.9), potential loss of top and/or bottom samples would not affect to much the slope of the line.

True, but the effect could be larger if much of the ice layer were melted/ablated away…

> Line 371-372: regardless of type of freezing, fractionation occurs. It is the type of freezing and fractionation that matters. In layers of ice formed by the freezing of thin films of water, all water freezes "at once" and as such the stable isotope composition of the resulting ice is similar to that of the water. In the case of freezing of a pool of water, fractionation and continuous incorporation of heavy isotopes in ice would result in stable isotope trend from top to the bottom. If the entire layer is sampled, the stable isotope composition of ice is similar to that of the parent water; but if samples are collected at various depths, their stable isotope composition would differ (higher values at top, lower at the bottom). Further, partial melting would result in loss of usually top layers, enriched in heavy isotopes, so that the stable isotope values of the ice would have little resemblance of that of original water. Elegantly, and nicely employed here, intersecting the LMWL with the slope of samples from every such layer would result in the original water before freezing (regardless of ice loss,

freezing intensity, fractionation factors etc). This intersection should be applied to layers with low values of the slope (2, 7, 8, 9, 11 and bottom) and one single data point per layer should be than plotted in fig. 8 (assuming thus that each of these layers formed from one single pool of water – as for example, the one in Fig. 10d). However, this would alter the discussion in lines 374-384.

As noted in Table 2, I made this correction to estimate the initial isotopic composition of the water forming layers 2, 4, 5, 7, 8, and 10, where slopes deviate markedly from 8. I did not initially correct layers 9, 11 and the bottom because their slopes were between 6.5 and 7, which is close to the local meteoric water line for winter precipitation. However, in the revision I will correct these values following the same protocol applied to the other layers. This adjusts the estimated "fSummer" slightly, but importantly it does not change the overall patter of increasing fSummer upward in the sampled ice exposure. The values measured and calculated for the different layers of ice are presented as the purple lines in Figure 8. Rather than a dot or symbol in the center of each layer, as you are suggesting here, I plotted a line spanning bottom to top in each layer to represent the estimated fSummer.

Line 380: I believe it is assumed here that the delta values reflect long-term annual means, in which case the difference is indeed large, pointing towards extreme climate shifts. Alternatively, it could be that they reflect different recharge patterns (e.g., winter vs. summer recharge).

This is exactly what I argue in the following paragraph (Lines 384-394).

Lines 379-384: in the absence of age control, I would refrain form discussing LGM-old ice. Further, it would be difficult to explain the survival of ice in a small cave, with such a dynamic ice accumulation/ablation processes.

I fully agree; my emphasis in this paragraph was on the unlikelihood that the observed changed in stable isotope values is entirely due to temperature. I'll reword to emphasize that it is very hard to imagine LGM-age ice persisting in this cave.

Lines 396-404: this could be somewhat shortened to a line, not to break the stable isotope discussion.

I would argue that it is important to present the details of this paragraph here because it supports my logic that the ice in WWC represents a snowpack signal augmented by varying amounts of summer precipitation, which is the focus of the following paragraphs.

Lines 404-412: perhaps a lengthier discussion of stable isotope variability in winter vs. summer could be included, with data from other stations nearby. Two data points are not enough to sustain the subsequent modeling. Further, if OIPC is used for summer, it should also be used for winter (especially as the OIPC-derived data for winter is depleted in 18O by about 3 ‰ compared to measured values). So, either used OIPC only, or data from stations, but not a combination of the two. I understand that this would affect the modeling in lines 412-420, but it is more correct.

True, two datapoints is not a large sample set. But as I note in the text, these are actual integrated snowpack samples from the elevation above the cave for the winter before the water was collected, not output from a model or interpolation scheme. Such correspondence gives these data value in excess of what would be

gained by resorting to the interpolated OPIC data. Shifting to a full reliance on the OPIC would change the results of the mixing model in terms of actual percents; but the key point in Figure 8, and the discussion in Lines 414-434, is that the relative abundance of winter and summer contributions to the ice in the cave changed over time. I am not trying to argue that a specific contribution was exactly a specific percent at a particular point in time, my focus is on the trends. Thus it seems supportable to use the snowpack data for the winter precip, and the OPIC for the summer.

> Lines 422-423: I still cannot see the reconstructed values.

I apologize, the way I wrote this was unclear; only the measured values are presented in Figure 8.

> Lines 422-434: see my comment above and perhaps redo the calculations (using one value for the layers formed as "lake ice")

I cannot use one value for all of the lake ice layers because they could have formed from water with differing mixtures of winter precip and summer rain.

> Lines 435-442: I find the discussion in this paragraph difficult to sustain by the data. Given the complex morphology of the ice surface and the lack of stratigraphic sketches, it is difficult to follow.

I hope to address this is the new figure I am planning, as discussed in my answer to your general comment about the chronology. Note that the stratigraphic depth corresponding to the dated organic remains is highlighted in Figure 8.

> Lines 444-456: in the absence of chronology, this section is very speculative and could be safely left out. "Glaciochemistry interpretation" should be moved before paleoclimate and "indication of recent change" after the climate data. However, in the absence of chronology, the entire discussion of the potential sources of variability in the chemistry of ice is speculative. As above, it could be reduced to a few lines, detailing the layer-by-layer variability, rather than potential temporal variability. For instance, are there differences between the two types of ice (lake vs. floor) in terms of chemistry? This could be more helpful for subsequent studies of cave ice deposits with better age control.

I would argue that the WWC record does not exist "in the absence of chronology": as I note above. I will further emphasize with a new figure, that the 14C dates can be used to constrain the age of this ice, and indicate that the lower levels of the ice mass accumulated during the Little Ice Age (as is currently highlighted in Figure 8). Thus, it seems prudent to carefully point out relevant records for snowfall and temperature in this region at this time. I am careful in Lines 444 -457 not to over-interpret the data. In my revision, I will add additional language noting that further age control is needed before these apparent connections can be rigorously evaluated.

> Line 469: see my comments on chronology. Ages should not be employed when dis- cussing the layers, as 1) it was only the surface that was dated and 2) complex surface morphology, with deep ridges.

As will be clarified in the new figure, the organic matter was dated from layers 7&8; thus it is defensible to assign a Little Ice Age age to the deeper ice (see Figure 8).

> "Limitations and direction for future research" and "conclusions" These two chapters should me merged and emphasize should be put on the potential usage of data obtained from this case in the general understanding of cave ice processes.

I disagree with the suggestion to merge these two sections. Acknowledging limitations is an important step for transparency, and suggesting directions for future work emphasizes the importance of this site and the foundational nature of this initial dataset. None of these are actual conclusions from the data analysis though; those are separate and should be presented in a separate section.

> The first panel in figure 2 is slightly misleading due to the inverted scale, please use a normal one (and perhaps write the "warmer outside, warmer in the cave" text with the same color as that of the line.

As is commonplace, the inverted scale was utilized to highlight the overall synchroneity between the different records. However, I will add note to the caption emphasizing that the temperature scale of this particular time series is inverted. I can also change the color of the text as you requested.

---

## Author Comment (AC2) · 2 Nov 2020

Thank you Marc for the obvious time and consideration you put into reviewing this manuscript. I appreciate your effort and am encouraged that you find this site, and the dataset I have generated, worthy of publication. As the title of the manuscript makes clear, this is a 'first investigation' of this recently discovered cave; the paper is not intended to be a complete summary of every type of investigation that could be conducted in such an environment. Mass balance of the ice mass, for instance, was not a target of this initial study. And given that nothing was known about the ice deposit (age, origin, composition) in this cave before this work, it would have been premature to use this location as a testbed for evaluating or answering key questions associated with ice caves. That work can – and should – come next, but the key goals here (as stated in Lines 65-67) are documenting the extent and age of the ice, elucidating its origin, and evaluating its paleoclimate potential. All of these goals are met by the data collected and presented in this study. In my revisions I will add wording to emphasize these specific goals, while also highlighting the potential of this site as a place where questions about ice caves could be addressed by future work building off the foundation presented here.

The author presents a detailed description of a recently discovered mid-latitude/high altitude ice cave located in the Uinta Mts, Utah (USA). The preliminary results suggest the cave ice is rather young and likely associated with enhanced winter snow precipitation during the Little Ice Age. Recent ingress of liquid water hints towards partial thawing of this sporadic mountain permafrost which is consistent with the general warming trend observed regionally.

The paper is nicely written and presents original data supporting the site-specific dynamics of the system. Such documentation is critical to better understand the response of the subsurface cryosphere with respect to climate change and provides fundamental data on a largely under-explored environment. Yet, the paper contributes only marginally to a better understanding of the general processes controlling cave ice mass balances nor does it provide a significant paleoenvironmental record. Therefore, I strongly suggest to emphasize the significance of this site for the larger community by stressing out its specificity with respect e.g. to other ice caves and/or its significance for regional mountain permafrost. Similarly, the discussion identifies a range of tasks (§4.5) that could be undertaken during future field work. Rather, I'd recommend focusing/rephrasing this chapter on what should be done at this site to answer key questions associated with ice caves, mountain permafrost and/or paleoenvironmental reconstructions. Providing these few adjustments and some minor comments below, I fully support publication of this paper.

Detailed comments:

These comments are all very helpful and I plan to incorporate them into my revisions. Thanks for giving the text such a close reading.

L. 16: edit "Most values of $\delta$18O and $\delta$D range plot subparallel" (delete "range")

L. 17: edit "with a slope of 7.5 with and an intercept at 0.03‰´' (delete "with")

L. 61-66: redundancy with l. 83-85.

L. 88: specify the location of the loggers

L. 90: what about the loggers' calibration?

L. 115: specify the location of your samples, in particular with respect to a cave ice stratigraphy and location of the isotope samples

L. 160: did you mean r2? Check significant digits

L. 194: can you comment about the wave propagation speed?

L. 219: I am missing some info about the modern context. What is the temperature mean and range, resp. the isotope signature, in the outside environment?

L. 220: "both isotopes are notably depleted" I guess you mean that both isotope ratios are low, i.e. depleted in the heavy isotopes (?)

L. 221-222: check significant digits on the isotope values

L. 272 "drops notably below ~0$^{\circ}$C" or below the cave air temperature? Fig. 8

check/remove unit of Mg/Ca

Table 1 I guess it is the sample mass and not the C-mass? Specify which calibration curve was used.

Table 2 check the significant digits; the precision is given at  0.2‰ and  1.0‰ for d18O and d2H, respectively (cf. l. 143)

---

## Author Response (AR1)

Over the past decade, investigations of perennial cave ice deposits took a more central stage, as the continuous rise in temperatures (and associated climatic changes) threaten these rather understudied components of the cryosphere. In this context, understanding the genesis and behavior of cave ice deposits could lead to the development of novel proxies of past climate variability that could add unique insights in past climate variability. Munroe attempts to do so by applying a range of investigation tools to several small perennial ice deposits in a cave in the Rocky Mountains, Utah and presenting and discussing the preliminary results. While the paleoclimatic potential of the investigated cave ice deposits is rather small, the results could advance our general knowledge of ice cave processes. While I had numerous comments, I nevertheless think that the manuscript could be published, given that 1) the paleoclimatic reconstruction is diminished in importance (see the comments on chronology) and 2) processes in the cave are emphasized as a tool for subsequent studies.

General comments: One of the main problems of the paper is the chronology. It is not clear at all what was actually dated. My understanding is that the top 15 cm of ice were dated. Because of the uneven ablation of the cave surface this approach is extremely problematic when it is intended to be used for climate reconstructions. If I understood correctly, the surface of the ice is extremely uneven and samples were collected from different depths, measured ("estimated visually" – a rather unusual choice) against the nearest ridge. Sublimation and/or melting was definitely acting with different intensities over the surface of the ice, bot today and in the past, and as such, younger packrat droppings could have been incorporated in older ice and or older ones reworked and deposited in new ice. Based on these considerations, I think the only message that can be obtained from the 14C dating is that ice was present in the cave between the youngest and oldest dates. Anyway, sampling locations need to be marked on the cave map. A detailed sketch of the ice surface with the position of the dated samples against morphology (and stratigraphy, if available) will help us understand what was being done.

I drafted a new figure (Figure 2) to display the stratigraphy of the ice, the relationship between the Frozen Freeway surface and the ultimate uppermost layer of the sampled ice exposure, and the stratigraphic level where the samples were collected for radiocarbon dating.

I am surprised that no attempt was made to collect and use bulk electromagnetic wave propagation velocities – these could have been used to peek into the composition of the ice (a method used by Hausmann & Behm, 2011, which is cited by the author). Is the data available and usable? Even if not "fantastic", it could further help subsequent studies (elsewhere). The point I want to make (here and through the review) is to have as much as possible methods descriptions and data available, rather than only "publishable" ones.

The suggestion that bulk EM wave propagation velocities be investigated is a good one. As I

note in the section on directions for future work, the investigations reported in this manuscript do not represent the end of fieldwork in this cave. Future geophysical investigations would certainly be useful for further clarifying three-dimensional architecture of the ice.

Specific comments The final paragraph of the introduction reads like being taken straight from a research proposal. Perhaps it should be rephrased to sound more article-like. Field site: it would benefit the readers to add one line about the characteristic of the limestone (rather than the name of the formation which would be in negligible interest). E.g., primary/secondary porosity and thickness of caprock are important to understand how water reaches the cave and/or how heat is being transferred to the cave. Also, for the understanding of ice chemistry, the general lithology of the rock should be presented.

I added details about the lithology of the Madison Limestone (Lines 72-73)

Line 89: at what height above the ice were the loggers suspended and at what distance from the rock walls? These are important considerations for the understanding of cave meteorology and factors leading to ablation/formation of ice, as the both presence of ice and of air currents induce strong vertical thermal gradients.

I added mention that the loggers were suspended 30 cm from the walls and ice (Line 91)

Line 100: please detail the "variety of gain settings" that were used during the GPR data acquiring (later in the manuscript, values are mentioned) and discuss the choice of one over the other. This is important if this study is to be useful for other researchers.

I clarified this in my revisions. (Lines 103-104)

Line 115-126: see my general comment. What calibration curve was used? The most recent is Raimer et al., 2020.

The IntCal13 curve was used, as is mentioned in Line 129. This was the newest available at the time the manuscript was written and there is very little difference in these young calibration ranges when recalibrating with Intcal 20.

Line 129-132: please describe the stratigraphy of the exposure and the number of samples collected per layer as the results are later presented using the layering. Did you consider layering during sampling or cut across strata (as suggested by the 2 cm spacing)? As water formed by the freezing of water, fractionation would have resulted in different stable isotope values within the same layer of ice.

Details about the stratigraphy of the exposure in the sampling strategy are presented in Lines 243 to 245, the visual divisions between the layers are shown schematically in the new Figure 2, the actual positions of these layer boundaries are plotted in Figure 8, and the number of samples per layer is stated in Table 2.

Line 134-136: please detail the location of the additional samples for stable isotope analyses.

These were added to Figure 1c.

Line 159: see my comment on the height of data loggers above cave ice surface.

Addressed in Line 91.

Line 167: using freezing degree days is rather uncommon, some readers might think that the number of days was calculated, rather than the sum of degrees below 0 °C. Please define it in the text. Perhaps the fdd should be calculated for the cave data, as well.

I defined the freezing degree day calculation. (Line 172)

Line 212-217: these should be moved under "methods"

Line 219-220: move to methods.

I'm not sure I agree with these two suggestions– talking about the total number of samples collected seems more like a result to me.

Line 220: "depleted" against. . .? Generally speaking, values cannot be depleted. A sample can be depleted in the heavy (light) isotope, resulting in a low (high) delta value.

Good point, I clarified throughout the text that delta values are relative to SMOW. I also switched to saying "less depleted" rather than "enriched", since all of these samples are depleted relative to SMOW>.

Line 224-229: I do not understand the reason for removing "outliers", the wide range of values is not an issue. This is a very unusual approach; I would consider all results, especially as the omitted results are almost similar to the ones used (e.g., fig. 7)

Screening to identify outliers in the isotope data seemed appropriate given the possibility that some of these ages have been greatly affected by fractionation.  In applying this approach I was following the work of Benjamin et al. (2005[1]), who screened their isotope data similarly before creating the local meteoric water line that I reference in my analysis. Perhaps this step is overly conservative, but I do think is important to evaluate the data before launching into analysis and interpretation.  In any event, only three of the samples were eliminated by this screening, so my analysis essentially moves forward with the data set intact.

Line 245: relative to. . .?

Changed to "less depleted" (Line 248)

Line 254-255: move to methods, and detail

This procedure is part of the analysis of these data, thus it seems more appropriate in the discussion.

Line 259: Ni – possible contamination of the upper layers?

I don't think so, because there is no source of Ni in the host rock.  And as for contamination in the laboratory, all of the samples are treated identically and all were run sequentially in the same batch on the ICP-MS.

Line 274-276: I do not understand the rationale behind the explanation for the chimney. If outside air gets below the internal one, cold air will flow inside the cave, regardless of the presence or absence of a chimney (perhaps references in lines 2776-277 should    be updated). Form the data in gig. 2, I do not see the need for a secondary entrance. The morphology of the cave and the data clearly indicates a "cold air trap", with dy- namic cooling as cold air flows inside the cave and slow warming, perhaps triggered by geothermal heat (and additional heat brought to the cave by dripwater). This does not exclude the presence of a chimney, but if it was not observed during the visits, perhaps the simplest explanation

[1]Benjamin, L., Knobel, L.L., Hall, L.F., Cecil, Ld., and Green, J.R., 2005, Development of a local meteoric water line for southeastern Idaho, western Wyoming, and southcentral Montana: http://pubs.usgs.gov/sir/2004/5126/.

would be sufficient (Occam's razor).

I added a bit more explanation for my logic in envisioning a chimney. I also clearly note that this is hypothetical. (Lines 277-280, again in Line 309)

Line 284-294: elegant

Thank you!

Line 292: did you notice airflow? Perhaps it is just warming propagating outwards though conduction from the warmer inner parts of the cave.

Yes, this is emphasized in the revision. (Line 279)

Line 295: what you mean by "back"? The longitudinal profile indicates flow from the entrance towards the "back" of the cave.

The observations in the summers of 2018 & 2019 clearly indicate that water entered from the rear of the cave, flooding out over the ice surface and reaching about halfway along the Frozen Freeway. I tried to clarify this in Lines 301-302 (It can't be shown on the scale of Figure 1, but the ice surface is slightly higher at the rear of the cave).

Line 299: again, was there a chimney/conduit observed? If not, water will just drip through the limestone's fissures.

I note that no drips or icicles were noted anywhere in the main part of the cave (Line 301)

Line 302: no need to flood the epikarst (it would be problematic at that altitude); in limestone, water will always find a way towards lower altitudes/caves.

True, I reworded this. (Lines 308-309)

Line 303: it would be useful to discuss the "ridges and troughs" on the surface of ice in detail. Inflow of warm water would definitely result in melting of ice, leading to the formation of surface micro topography. Further, the same water could bring in young packrat droppings and redeposit them in older ice, exposed during melting. This is evident in figure 10, where ridges are drowned in newly formed ice.

There is no sign that the ridges and troughs are produced by warm water, rather they appear to have been sculpted by airflow. Certainly water flowing over the surface of the ice could we distribute cryogenic minerals and organic matter, but it would not redeposit this material within "older ice"; rather this process would create a situation where older material is encased in younger ice. That material would be concentrated in a single stratigraphic level, clearly indicating its origins through melt out related concentration. That is not the case with the organic matter dated in this study, as I noted above.

Line 304: not seeing a secondary entrance, only hypothesize one, I would consider the cave a "static cave with congelation ice" (Luetscher and Jeannin, 2004a)

Again, given the clear evidence for strong airflow through the cave in the form of the sculpted ice surface, and the fact that strong drafting was noted during each field visit, I am inclined to lean toward the interpretation of hypothesized but currently inaccessible second entrance. Nonetheless, I am careful to note the hypothesized nature of this conduit. (Line 309)

Line 318: Is there possibly ice present under the breakdowns (in the "ice free" sections)?

Perhaps a continuous layer of ice extends from the entrance through the cave and the breakdowns cover it in places. It would be otherwise difficult to understand the lack of ice in parts of the cave.

Possibly, but this cannot be proven or disproven from the available evidence.  In reality I think that the central and forward part of the cave where the breakdown is concentrated probably represent a mixing zone between ice derived from water entering the rear of the cave and flowing forward, and ice derived from water that runs into the Icicle Room through the entrance.

Line 324-326: the low reflectivity could indicate a thick and homogenous layer of clear ice, formed by the slow freezing of water in a through between ridges (so-called "lake ice").  And indeed, this ice would be free of cryogenic calcite and other sediments,   that would settle at the bottom of the lake water during freezing. Here the usage of use bulk electromagnetic wave propagation velocities could help. Line 334: a photo  and stratigraphic sketch would help understand the structure of the ice deposit and the stable isotope values

This is a good point, however looking at the current surface of the ice where it has been sculpted by the airflow reveals a nearly continuous layer of concentrated cryogenic minerals.  These would form a strong reflector in GPR data.  Furthermore, the scale of these ridges and troughs is on the order of 10 to 30 cm, whereas the scale of the apparently reflector-free ice seen in the GPR data is on the order of the meter.  If continuous, meter deep pools of water were present on the surface of the ice at some point in the past, then yes they could have frozen as a lake ice creating a GPR record like the one that I imaged.  However, this would require ridges and troughs on a scale quite different than that observed in the 2018 ice surface.

Line 342: depleted in heavy isotopes

I made this change in Line 402

Line 363: technically, it was the isotopologues that were fractionated. . .

Line 364-366: because during freezing of a pool of water the samples align one a straight line in a dH-$\delta$18O diagram (with r2>0.9), potential loss of top and/or bottom samples would not affect to much the slope of the line.

True, but the effect could be larger if much of the ice layer were melted/ablated away…

Line 353 and subsequent paragraph: "freezing slopes" have been described as being generally below 7.2 (Jouzel and Souchez, 1982, Souchez and Jouzel, 1984, Souzhez et al., 2000, Persoiu et al., 2011). Layers 4, 5, 6, 10, 12 and 13 (but not 9 and 10, which have slopes below 7) likely formed as thin layers of water froze on top of the existing ice block (slope>7). This could be checked by plotting d-exces vs. d2H (Souzchez et al., 2000) for every layer. Lack of correlation between the two parameters would indicate kinetic conditions and thus open-system freezing (thin layers of water freezing on top of existing ice). Alternatively, all layers could have formed as "lake ice" and subsequently part of them melted away thus resulting in the loss of the alignment along a line with a slope below 7.

Line 371-372: regardless of type of freezing, fractionation occurs. It is the type of freezing and fractionation that matters. In layers of ice formed by the freezing of thin films of water, all water freezes "at once" and as such the stable isotope composition of the resulting ice is similar to that of the water. In the case of freezing of a pool of water, fractionation and continuous incorporation of heavy isotopes in ice would result in stable isotope trend from top to the bottom. If the entire layer is sampled, the stable isotope composition of ice is similar to that of the parent water; but if samples are collected at various depths, their stable isotope composition would differ (higher values at top, lower at the bottom). Further, partial melting would result in loss of usually top layers, enriched in heavy isotopes, so that the stable isotope values of the ice would have little resemblance of that of original water. Elegantly, and nicely employed here, intersecting the LMWL with the slope of samples from every such layer would result in the original water before freezing (regardless of ice loss,

freezing intensity, fractionation factors etc). This intersection should be applied to layers with low values of the slope (2, 7, 8, 9, 11 and bottom) and one single data point per layer should be than plotted in fig. 8 (assuming thus that each of these layers formed from one single pool of water – as for example, the one in Fig. 10d). However, this would alter the discussion in lines 374-384.

As noted in Table 2, I made this correction to estimate the initial isotopic composition of the water forming layers 2, 4, 5, 7, 8, and 10, where slopes deviate markedly from 8.  I did not initially correct layers 9, 11 and the bottom because their slopes were between 6.5 and 7, which is close to the local meteoric water line for winter precipitation.  However, in the revision I have now corrected these values following the same protocol applied to the other layers.  This adjusts the estimated "fSummer" slightly, but importantly it does not change the overall patter of increasing fSummer upward in the sampled ice exposure.  The values measured and calculated for the different layers of ice are presented as the purple lines in Figure 9.  Rather than a dot or symbol in the center of each layer, as you suggest, I plot a line spanning bottom to top in each layer to represent the estimated fSummer.  (Lines 416-422 and a new Table 2).

Line 380: I believe it is assumed here that the delta values reflect long-term annual means, in which case the difference is indeed large, pointing towards extreme climate shifts. Alternatively, it could be that they reflect different recharge patterns (e.g., winter vs. summer recharge).

This is exactly what I argue in Lines 443-444.

Lines 379-384: in the absence of age control, I would refrain form discussing LGM-old ice. Further, it would be difficult to explain the survival of ice in a small cave, with such a dynamic ice accumulation/ablation processes.

I fully agree; my emphasis in this paragraph was on the unlikelihood that the observed changed in stable isotope values is entirely due to temperature.  I'll reword to emphasize that it is very hard to imagine LGM-age ice persisting in this cave. (Lines 350-352).

Lines 396-404: this could be somewhat shortened to a line, not to break the stable isotope discussion.

I would argue that it is important to present the details of this paragraph here because it supports my logic that the ice in WWC represents a snowpack signal augmented by varying amounts of summer precipitation, which is the focus of the following paragraphs.

Lines 404-412: perhaps a lengthier discussion of stable isotope variability in winter vs. summer could be included, with data from other stations nearby. Two data points are not enough to sustain the subsequent modeling.  Further, if OIPC is used for summer,  it should also be used for winter (especially as the OIPC-derived data for winter is depleted in 18O by about 3 ‰ compared to measured values). So, either used OIPC only, or data from stations, but not a combination of the two. I understand that this would affect the modeling in lines 412-420, but it is more correct.

True, two datapoints is not a large sample set.  But as I note in the text, these are actual integrated snowpack samples from the elevation above the cave for the winter before the water was collected, not output from a model or interpolation

scheme (Lines 374-380). Such correspondence gives these data value in excess of what would be gained by resorting to the interpolated OPIC data. Shifting to a full reliance on the OPIC would change the results of the mixing model in terms of actual percents; but the key point in Figure 9, and the discussion, is that the relative abundance of winter and summer contributions to the ice in the cave changed over time. I am not trying to argue that a specific contribution was exactly a specific percent at a particular point in time, my focus is on the trends. Thus it seems supportable to use the snowpack data for the winter precip, and the OPIC for the summer.

> Lines 422-423: I still cannot see the reconstructed values.

I apologize, the way I wrote this was unclear; only the measured values are presented in Figure 9.

> Lines 422-434: see my comment above and perhaps redo the calculations (using one value for the layers formed as "lake ice")

I cannot use one value for all of the lake ice layers because they could have formed from water with differing mixtures of winter precip and summer rain.

> Lines 435-442: I find the discussion in this paragraph difficult to sustain by the data. Given the complex morphology of the ice surface and the lack of stratigraphic sketches, it is difficult to follow.

I address this with the new Figure 2. Note that the stratigraphic depth corresponding to the dated organic remains is highlighted in Figure 9.

> Lines 444-456: in the absence of chronology, this section is very speculative and could be safely left out. "Glaciochemistry interpretation" should be moved before paleoclimate and "indication of recent change" after the climate data. However, in the absence of chronology, the entire discussion of the potential sources of variability in the chemistry of ice is speculative. As above, it could be reduced to a few lines, detailing the layer-by-layer variability, rather than potential temporal variability. For instance, are there differences between the two types of ice (lake vs. floor) in terms of chemistry? This could be more helpful for subsequent studies of cave ice deposits with better age control.

I would argue that the WWC record does not exist "in the absence of chronology": as I note above. I emphasize this with the new Figure 2, illustrating that the $^{14}$C dates can be used to constrain the age of this ice, and indicate that the lower levels of the ice mass accumulated before and during the Little Ice Age (as is also highlighted in Figure 9). Thus, it seems prudent to carefully point out relevant records for snowfall and temperature in this region at this time. I am careful in Lines 455-466 not to over-interpret the data.

> Line 469: see my comments on chronology. Ages should not be employed when dis- cussing the layers, as 1) it was only the surface that was dated and 2) complex surface morphology, with deep ridges.

As is clarified in the new Figure 2, the organic matter was dated from layers 7 & 8; thus it is defensible to assign a Little Ice Age age to the deeper ice (see Figure 9).

> "Limitations and direction for future research" and "conclusions" These two chapters should me merged and emphasize should be put on the potential usage of data obtained from this case in the general understanding of cave ice processes.

I disagree with the suggestion to merge these two sections. Acknowledging limitations is an important step for transparency, and suggesting directions for

future work emphasizes the importance of this site and the foundational nature of this initial dataset. None of these are actual conclusions from the data analysis though; those are separate and should be presented in a separate section.

> The first panel in figure 2 is slightly misleading due to the inverted scale, please use a normal one (and perhaps write the "warmer outside, warmer in the cave" text with the same color as that of the line.

As is commonplace, the inverted scale was utilized to highlight the overall synchroneity between the different records. However, I added note to the caption emphasizing that the temperature scale of this particular time series is inverted (Line 757). I also change the color of the text in the figure as you requested.

The Cryosphere Discuss., https://doi.org/10.5194/tc-2020-152-RC2, 2020

[Figure]

Thank you Marc for the obvious time and consideration you put into reviewing this manuscript. I appreciate your effort and am encouraged that you find this site, and the dataset I have generated, worthy of publication. As the title of the manuscript makes clear, this is a 'first investigation' of this recently discovered cave; the paper is not intended to be a complete summary of every type of investigation that could be conducted in such an environment. Mass balance of the ice mass, for instance, was not a target of this initial study. And given that nothing was known about the ice deposit (age, origin, composition) in this cave before this work, it would have been premature to use this location as a testbed for evaluating or answering key questions associated with ice caves. That work can – and should – come next, but the key goals here (as stated in Lines 65-67) are documenting the extent and age of the ice, elucidating its origin, and evaluating its paleoclimate potential. All of these goals are met by the data collected and presented in this study. In my revisions I will add wording to emphasize these specific goals, while also highlighting the potential of this site as a place where questions about ice caves could be addressed by future work building off the foundation presented here.

The author presents a detailed description of a recently discovered mid-latitude/high altitude ice cave located in the Uinta Mts, Utah (USA). The preliminary results suggest the cave ice is rather young and likely associated with enhanced winter snow precipitation during the Little Ice Age. Recent ingress of liquid water hints towards partial thawing of this sporadic mountain permafrost which is consistent with the general warming trend observed regionally.

The paper is nicely written and presents original data supporting the site-specific dynamics of the system. Such documentation is critical to better understand the response of the subsurface cryosphere with respect to climate change and provides fundamental data on a largely under-explored environment. Yet, the paper contributes only marginally to a better understanding of the general processes controlling cave ice mass balances nor does it provide a significant paleoenvironmental record. Therefore, I strongly suggest to emphasize the significance of this site for the larger community by stressing out its specificity with respect e.g. to other ice caves and/or its significance for regional mountain permafrost. Similarly, the discussion identifies a range of tasks (§4.5) that could be undertaken during future field work. Rather, I'd recommend focusing/rephrasing this chapter on what should be done at this site to answer key questions associated with ice caves, mountain permafrost and/or paleoenvironmental reconstructions.

Providing these few adjustments and some minor comments below, I fully support publication of this paper.

Detailed comments:

L. 16: edit "Most values of $\delta$18O and $\delta$D range plot subparallel" (delete "range")

Done (Line 16)

L. 17: edit "with a slope of 7.5 with and an intercept at 0.03‰´' (delete "with")

Done (Line 17)

L. 61-66: redundancy with l. 83-85.

In Line 83-85 (now 85-87) I am specifying what was done in the field.

L. 88: specify the location of the loggers

These are noted in Figure 1c. I added mention of how far away they were suspended from the ice and cave walls (Line 91).

L. 90: what about the loggers' calibration?

They are factory calibrated and were cross-checked before deployment.

L. 115: specify the location of your samples, in particular with respect to a cave ice stratigraphy and location of the isotope samples

I added these to Figure 1c and the caption (Lines 737-741).

L. 160: did you mean r2? Check significant digits

Yes, thanks you (Line 65).

L. 194: can you comment about the wave propagation speed?

Added to Line 112.

L. 219: I am missing some info about the modern context. What is the temperature mean and range, resp. the isotope signature, in the outside environment?

Outside temperature data are presented in Lines 168-174; snow and precip isotope values in Lines 360-362 and 374-378.

L. 220: "both isotopes are notably depleted" I guess you mean that both isotope ratios are low, i.e. depleted in the heavy isotopes (?)

I added relative to SMOW (Line 223).

L. 221-222: check significant digits on the isotope values

I adjusted these throughout.

L. 272 "drops notably below ~0$^{\circ}$C" or below the cave air temperature?

Good point, I meant the outside temperature – but because the inside never gets above 0, it's really the same.

Fig. 8 check/remove unit of Mg/Ca

Because it not really an important part of the discussion, I removed the Mg/Ca curve from Figure 9 entirely.

Table 1 I guess it is the sample mass and not the C-mass? Specify which calibration curve was used.

The sample curve is mentioned in the text (Line 29 and 789).

Table 2 check the significant digits; the precision is given at 0.2‰ and $\pm$ 1.0‰ for d18O and d2H, respectively (cf. l. 143)

Good point, I made this change throughout the text and in Table 2.

[revised manuscript text omitted]

---

## Author Response (AR2)

I appreciate the careful reading by Editor Hauck and the thoughtful comments from Dr. Perşoiu. The manuscript has certainly been improved by incorporation of their feedback.

In response to the brief comments from Dr. Hauck, changes were made as follows (line numbers refer to the document with changes accepted):

line 9: "Temperatures ...were not recorded..." (insert s ?)

This change was made on Line 9.

line 327 "of a breakdown" ?

-I did not make this change. Cave "breakdown" is commonly used as a noun describing, rubble that has fallen from the cave ceiling.

line 470: "did not change" ?

-I made this change (Line 471).

line 515: "observations ...suggest"

-I made this change (Line 516).

Caption Figure 6: insert "survey" after "Results of an 8-m long Ground Penetrating Radar (GPR)"

-I made this change (Line 778).

In response to the helpful comments from Aurel Perşoiu:

Please add info to the type and frequency of porosity (primary and secondary, including information on the presence of fractures and fissures). These info are important for the understanding of water and air circulation within the rock above the cave and ultimately on how water reaches the cave to freeze into the ice block

> -Unfortunately, not much is known about the fracture pattern and porosity in the host rock beyond what can be observed in the cave itself. I added some information to the cave description to provide a bit of additional detail about these fractures though (Line 79-80).

14C Calibration – please use the newest curve (IntCal 20). The differences are small compared to IntCal13, indeed, but than, why bother with advancements if we don't use them?

> -Point well taken. I recalibrated the $^{14}$C results with IntCal 20 in Oxcal, updated Table 1, adjusted the text accordingly, and remade Figure 7.

Stable isotope nomenclature
E.g, lines 18-20 in the abstract: In line 18, δ18O and δ2H values are discussed and than in line 19 "depleted winter precipitation" are mentioned. Winter precipitation can be depleted in 18O relative to summer precipitation, not in δ18O. In general, depleted, enriched, less depleted etc should be avoided – being comparative words, every time the two terms of the comparison should be presented. This depletion does not usually refers to SMOW, as except for some highly evaporated samples or os, all others are depleted relative to SMOW. Please say something like "winter precipitated is depleted in heavy isotopes compared to summer one", "winter precipitations has lower δ18O values", "winter pp has lower 18O/16O ratios compared to summer one" or similar throughout the manuscript. Check also Sharp, Z. Principles on stable isotope geochemistry (2nd edition)

> - I understand your point and appreciate your help in improving the writing. I made numerous changes throughout to eliminate comments like "depleted winter precipitation". When I do mention depletion, I added mention of the benchmark to which depletion is relative. (e.g. Lines 351, 362, 363, 383, 469, etc.)

This should be mentioned in the article: "Possibly, but this cannot be proven or disproven from the available evidence. In reality I think that the central and forward part of the cave where the breakdown is concentrated probably represent a mixing zone between ice derived from water entering the rear of the cave and flowing forward, and ice derived from water that runs into the Icicle Room through the entrance."

-What was perhaps not clear in previous versions of the manuscript is the fact that the breakdown is along a slope; it is not a flat surface of ice mantled by rubble. This was always presented in Figure 2 (the cave profile), but I added additional language to the cave description to clarify this (Lines 80-81).

Line 382: please detail the mixing model and how the quoted percentage were obtained

– I updated the text to clarify that this was a simple linear mixing model between two end members, representing winter and summer precipitation (Lines 383-384).